

# Wave activity in front of high-$\beta$ Earth bow shocks

Anatoli A. Petrukovich[1], Olga M. Chugunova[1], and Pavel I. Shustov[1]

[1]Space Research Institute of Russian Academy of Sciences, Moscow, Russia

**Correspondence:** A.A.Petrukovich (a.petrukovich@cosmos.ru)

**Abstract.** Earth's bow shock in high $\beta$ (ratio of thermal to magnetic pressure) solar wind environment is relatively rare phenomenon. However such a plasma object may be of interest for astrophysics. We survey statistics of high-$\beta$ ($\beta > 10$) shock observations by near-Earth spacecraft since 1995. Typical solar wind parameters related with high $\beta$ are: low speed, high density and very low IMF 1–2 nT. These conditions are usually quite transient and need to be verified immediately upstream of

the observed shock crossings. About a hundred crossings were initially identified mostly with quazi-perpendicular geometry and high Mach number. In this report 22 Cluster project crossings are studied with spacecraft separation within 30–200 km. Observed shock front structure is different from that for quaziperpendicular supercritical shocks with $\beta \sim 1$. There is no well defined ramp. Dominating magnetic waves have frequency 0.1–0.5 Hz (in some events 1–2 Hz). Polarization has no stable phase and is closer to linear. In some cases it is possible to determine wavelength at 0.1–0.5 Hz of the order of 200–900 km.

*Copyright statement.*

## 1 Introduction

Shocks are the primary dissipation mechanism in space plasmas with supersonic flows (Sagdeev, 1966; Kennel et al., 1985; Krasnoselskikh et al., 2013). A brand new branch of plasma science, theory of collisionless shocks, appeared in the sixties, in response to new observational data on solar flares and solar wind interaction with Earth magnetic field.

In the solar system solar wind forms the bow shocks at planets and comets, the termination shock at the heliospheric interface. Interplanetary shocks develop, when large-scale transient structures propagate in solar wind after solar eruptions. In the distant space, shocks are associated with supernova explosions, stellar winds, collisions of galaxy clusters. Astrophysical shocks are believed to have a leading role in the acceleration process of cosmic rays (Krymskii, 1977; Axford et al., 1977). The review of space shock physics can be found in AGU Geophysical Monographs 34 and 35 (1985). The Earth bow shock has been most

thoroughly studied since the launch of the first spacecraft and is the main source of our in-situ knowledge of collisionless shock structure and dynamics.

Of particular interest to astrophysical applications are shocks in weak magnetic field environment (high-$\beta$ shocks) (e.g., Markevitch and Vikhlinin, 2007). $\beta$ is a dimensionless parameter, a ratio of plasma thermal to magnetic energy density. Unfortunately, observations of high $\beta$ shocks near the Earth are quite rare, since the solar wind plasma usually has $\beta \sim 1$. Very



few such investigations were published, merely checking validity of Rankine-Hugoniot conditions and marking high amplitude of magnetic variations in the front (Formisano et al., 1975; Winterhalter and Kivelson, 1988; Farris et al., 1992). Note, that is some investigations moderate $\beta \geq 1$ is described as "high-$\beta$" regime (e.g., $\beta = 2.4$ in Scudder et al., 1986).

Electromagnetic fields and waves in space shocks are of primary importance, since in the absence of collisions, kinetic mech-

anisms of field-particle interactions are responsible for dissipation and particle acceleration (Sagdeev, 1966; Krasnoselskikh et al., 2013). Of particular interest are relatively low frequency waves, which visually have maximal amplitudes, since they actually form the shock front structure, dissipating ions. A number of turbulence theories were also suggested for the high-$\beta$ shocks (Kennel and Sagdeev, 1967a, b; Coroniti, 1970). Due to presence of magnetic field a wide variety of shock types exists with quite differing structure (Kennel et al., 1985).

For example, in a supercritical quazi-perpendicular shock, the oblique whistler waves near lower-hybrid frequency ($\sim$5 Hz) form the ramp (sharp jump of magnetic field) via the non-linear steepening and decay cycle (Krasnoselskikh et al., 2002, and references therein). In several studies the wavelength of these waves and the scale of shock ramp were determined to be around 10-s of km and oscillations were in fact identified as whistlers (Petrukovich et al., 1998; Walker et al., 2004; Hobara et al., 2010; Schwartz et al., 2011; Dimmock et al., 2013; Krasnoselskikh et al., 2013) Another issue of interest is electron heat-

ing, which requires sufficiently small scale variations for non-adiabatic (transverse) acceleration and following isotropisation (Balikhin et al., 1993; Vasko et al., 2018).

To approach the study of magnetic structures in high-$\beta$ shocks with the Earth's bow shock observations we scanned the whole set of available spacecraft data. We start with the general occurrence statistics of high-$\beta$ solar wind and then look into some cases of multipoint observations allowing to estimate spatial wave characteristics. We use $\beta > 10$ criterion, which is

justified in the course of this presentation.

Solar wind and IMF data were taken from OMNI-2 data set, the 1-hour variant was used for the initial survey, and the 1-min variant — for final categorization of crossings. $\beta$ values are precalculated in OMNI-2, assuming constant electron temperature, He++ fraction and He++ temperature. To access possible solar wind variability we use also ACE and Wind final Earth-shifted data from OMNI archive.

The period of analysis was 1995–2017, which has almost full coverage of interplanetary measurements and many spacecraft crossing bow shock. We used Interball (1995–2000), Geotail (since 1995), Cluster (since 2000) and THEMIS (since 2007) orbital and spin-averaged magnetic field data from CDAWeb archive. For the detailed analysis we used full-resolution Cluster FGM magnetic field (Balogh et al., 2001) and HIA/CODIF ion data (Rème et al., 2001) from Cluster Final Archive. All vectors are in GSE frame of reference.

## 2  Solar wind statistics and details of search procedure

We used 1-hour OMNI data for the period 1995–2017 to determine the occurrence of high $\beta$ solar wind. The average solar wind $\beta$ is somewhat large than unity. High $\beta$ conditions are unevenly distributed across solar cycles (Fig. 1), being more frequent at



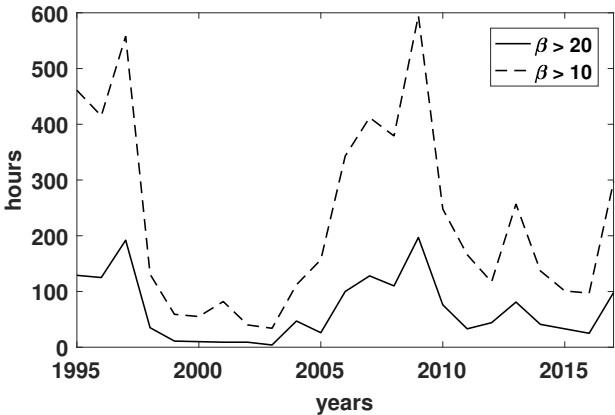

**Figure 1.** Number of hours with high $\beta$ with respect to calendar year.

the solar minima 1996–1997 and 2007–2009. For the threshold $\beta > 10$ there are 50–500 hours per year, while for $\beta > 20$, the number is about 3–5 times smaller.

Figure 2 shows distributions of magnetic field magnitude, solar wind speed, density and total static pressure for the full dataset of one-hour values during 1995–2017 and for the subset $\beta > 10$. The high $\beta$ corresponds to slow, cold and dense

solar wind with low magnetic field (ion temperature not shown here). However total static (magnetic plus thermal) pressure distribution is similar (Fig. 2b). Thus the high-$\beta$ events are mostly depressions of magnetic field, compensated (at least on statistics) by increase of plasma density. The only notable difference of distributions for $\beta > 20$ (Fig. 2a, red line) is more frequent presence of magnetic field $\sim$1 nT, with the average 1.6 nT, while for $\beta > 10$ the average is $\sim$2.2 nT.

According to Fig.3 more than 50% of events with $\beta > 10$ have one-hour duration (one point in the analyzed OMNI variant).

A sample event is in Fig. 4. There is one-hour long decrease of magnetic field and density increase, corresponding to $\beta$ rise to about 20. At an occasional depletion of magnetic field below 2 nT $\beta$ jumps to about 40–80 for few minutes.

Since formation of high $\beta$ conditions mostly depends on subtle variations of magnetic field magnitude around 1–2 nT (note, that $\beta$ has square dependence on magnetic field), it should be quite sensitive to spatial inhomogeneity of solar wind and IMF, and, in particular, to differences between those detected at L1 (in OMNI dataset) and actually hitting Earth. Fig. 5

shows comparison of $\beta$ calculation for Wind and ACE 1-hour data (only for times, when Wind data were used in OMNI). The scatter is quite large. Thus actual $\beta$ conditions need to be rechecked with local measurements. This issue is elaborated more in Discussion.

The semi-automated algorithm was used to assemble initial statistics of the shock candidates. For each 1-hour point in OMNI with $\beta > 10$, we checked for possible spacecraft location within 5 $R_E$ from the model bow shock (Farris et al., 1991). In a case

any spacecraft was in place, the plots of solar wind, IMF, local magnetic field and plasma parameters were analyzed visually in the 5-hour window around the selected hour. Broad temporal and spatial spans were used to ensure that all possible crossings of a moving bow shock are captured for future analysis. Only events with clear shock traversals (jumps in magnetic field and



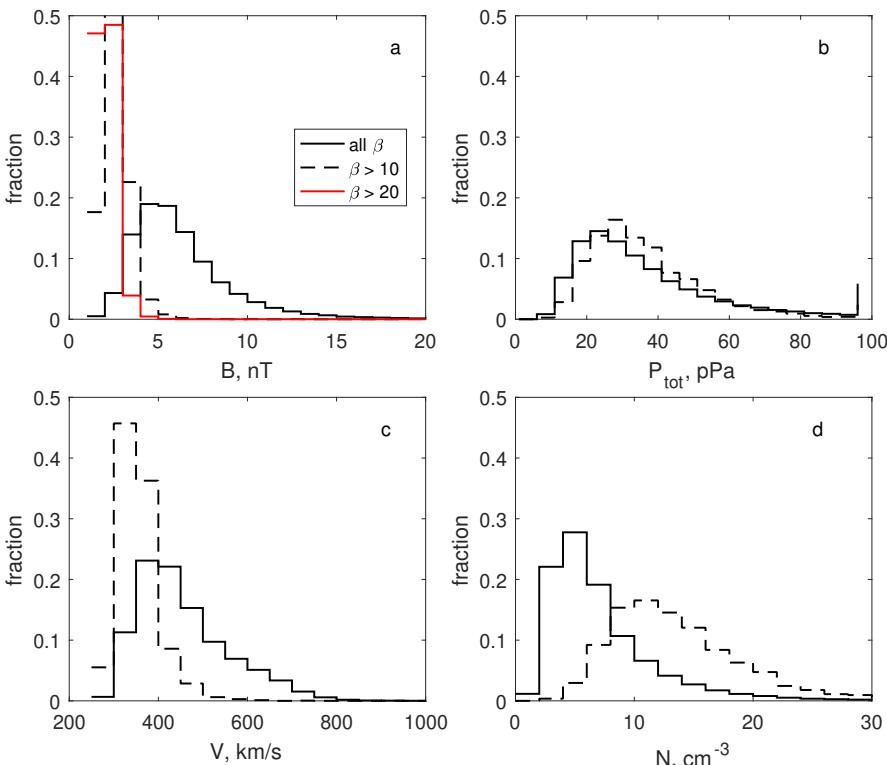

**Figure 2.** Histograms of solar wind and IMF occurrence for 1995–2017 (solid lines) and for $\beta > 10$ (dashed lines) subset. (a) Total magnetic field (red line corresponds to $\beta > 20$) , (b) total static pressure, (c) solar wind speed, (d) ion density.

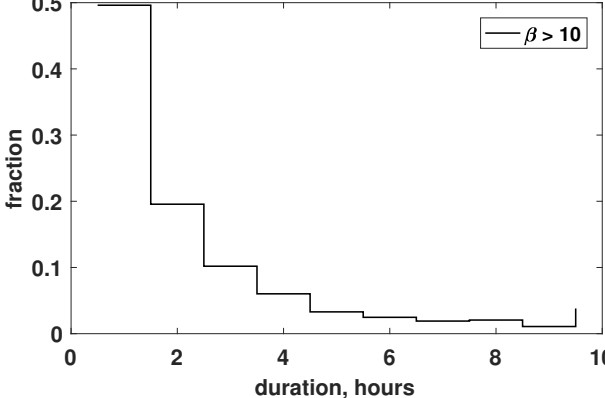

**Figure 3.** Histogram of duration of $\beta > 10$ intervals in round hours.

ion density) were accepted. Such manual selection has definite bias to quasi-perpendicular shocks (which usually have a clear





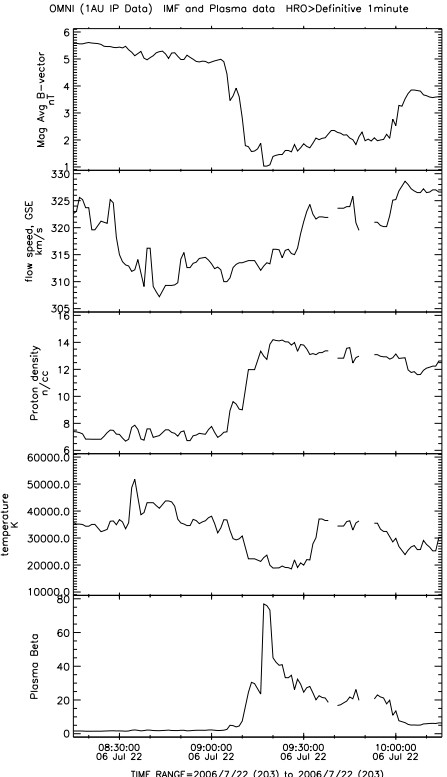

**Figure 4.** Example of high-$\beta$ interval. From top to down: magnetic field magnitude, solar wind speed, proton density, proton temperature, plasma $\beta$. 1-min OMNI data set used.

step-like front), but it was considered acceptable for this particular study. The most of these initially selected intervals actually contained no shock crossings.

Actual $\beta$ at the particular shock crossings were checked with 1-min OMNI data. It was often below 10, either because registered shocks were just outside initially selected hours, or because $\beta$ varied on a time scale, smaller than an hour. Since a change of $\beta$ is usually related with the solar wind density change, it is associated also with the dynamic pressure change. The latter drives a large-scale shock motion and probability of shock registration by a spacecraft increases. In fact, many shock crossings were registered at a boundary of $\beta$ change and such events were also discarded, since it was impossible to attribute them to stable plasma conditions.

Finally the list contained about a 100 individual crossings with average $\beta$ about 20 (1-min value at shock front crossing). The choice of initial threshold $\beta > 10$ (for 1-hour points) was finally justified at this stage, since a variant with initial $\beta > 20$




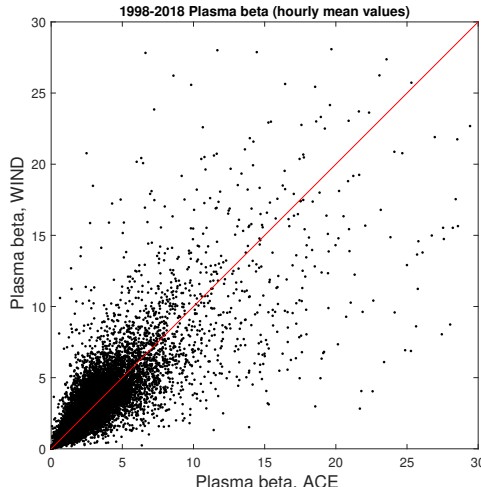

**Figure 5.** Comparison of Wind and ACE $\beta$ using 1-hour data. See text for details. Red line is bisector

resulted with almost empty final list. However, all these events still need a more detailed confirmation, in particular, of local high $\beta$, stable enough crossing velocity, plasma data availability etc.

For the specific analysis in this investigation we selected 22 Cluster project crossings with relatively small spacecraft separation. One event is from 2003, with the Cluster tetrahedron size of about 300 km, while the other are for the late years
2008–2016, when separation between C3 and C4 was 30–200 km (Table S1 in Supplement 1). Some of these examples are presented below.

## 3   Shock examples

### 3.1   Event 1

The first example was registered by Cluster C3 and C4 spacecraft on 18 December 2011 (1436–1440 UT) with the spacecraft
separation 36 km. Solar wind speed was small ∼260 km/s, IMF magnitude — 2.5 nT (all characteristics are in Table S1). The model shock normal angle with respect to IMF was $46^o$ (using Farris et al. (1991) model). Spacecraft orbit is almost parallel to the model shock front (Fig. 6), but shock velocity is definitely much higher than the spacecraft velocity. Alfven Mach number is ≈18, magnetosonic Mach number is ≈5, current $\beta$ (according to 1-min OMNI) is 10.8. Thus this is quazi-perpendicular supercritical bow shock, which structure for standard $\beta$ is well studied (Scudder et al., 1986; Krasnoselskikh et al., 2013).
Fig. 7 contains overview of magnetic field and plasma parameters. The shock front is somewhat arbitrarily placed at 14:37:45 UT (marked by vertical line) at a first extended peak of magnetic field. The shock foot, the zone with reflected ions (Fig. 7f), gradual increase of density and magnetic field, maximum of parallel ion temperature, is at 14:37:45–14:40:40 UT. The detailed





analysis of ion dynamics will be performed elsewhere. Solar wind magnetic field measured locally by Cluster is the same as OMNI data (compare two lines in Fig. 7d), therefore OMNI $\beta$ value is confirmed.

The final value of downstream magnetic field is around 10 nT, and compression ratio is thus close to maximally possible value of 4, in accordance with the high Mach number. However, the observed front structure is very different in comparison with

5 that expected for a supercritical shock. First of all, there is no well-defined magnetic field jump (magnetic ramp). Thus it proved to be impossible to determine reliably shock speed and spatial scale of shock transition. The magnetic field increase is wavy rather than step-like, magnetic magnitude is often down to 5 nT. Second, there is a zone upstream from the largest magnetic bursts with slowly increasing magnetic field and density almost up to the expected downstream value (14:37:45–14:38:20).

We highlight interval 14:37:00–14:38:30 (Fig. 8) as an example of wave activity. Frequency spectra are in Fig. 9. The

10 magnetic profile is dominated by a wave with frequency around 0.3 Hz and amplitude up to 20 nT, more pronounced in $B_y$ and $B_x$ components. An interval 14:37:27–14:37:47 of the most intense oscillation is taken to estimate the wavelength. The main oscillation (0.3 Hz) is very similar at two spacecraft and visually the time shift between C3 and C4 is about a fraction of a second.

Parameters of waves, filtered in frequency range 0.1–0.77 Hz, are in Table 1. Vector of maximum variance is almost along

local magnetic field ($B_y$ component dominates), of minimum variance — along $Z$. Ratios of eigenvalues are $\lambda_{min}/\lambda_{int} = 0.34$, $\lambda_{int}/\lambda_{max} = 0.58$, and one may assume elliptic polarisation with relatively defined propagation direction. The time shift between magnetic measurements along the maximum variance component is 0.13 s (determined with correlation analysis), while the spacecraft separation along the minimum variance direction is 10 km. The resulting wavelength estimate is 250 km. However the hodograph of magnetic field rotation (Fig. 10) shows that the polarization actually might be linear with variable

direction. In such a case propagation direction is undefined. The maximum possible wavelength ∼900 km can be obtained taking maximum possible separation 36 km. The estimate of the Doppler shift can be obtained taking either full local proton velocity 146 km/s, or its projection to minimal eigenvector 41 km/s and is 0.04–0.58 Hz, depending also on a variant of the wavelength estimate.

Finally we note the oscillations with higher frequency about 1 Hz and smaller amplitude of couple nT, which are best

observable in $B_z$ component (Fig. 8c and Fig. 9). These oscillations are quite different at two spacecraft and the wavelength analysis proved to be not possible. The eigenvalue ratios (after filtering the frequency range 0.7–10 Hz) are $\lambda_{min}/\lambda_{int} = 0.68$, $\lambda_{int}/\lambda_{max} = 0.49$, thus reliable determination of any wave proper direction is definitely not possible.

## 3.2 Other events

In this subsection we briefly present two shock examples with substantially different wave activity. A shock from January

4th, 2008 (1600–1604 UT) was registered with Cluster separation about 40 km, and very similar solar wind conditions (Table S1, Fig.S1 in Supplement). The detailed wave activity at the front is presented in Fig. 11. General frequency structure of waves in this event is similar to that in Example 1. There is a dominating oscillation with frequency about 0.4–0.5 Hz, as well as the lower amplitude waves with frequency above 1 Hz. The specific feature is strong difference of C3 and C4 oscillation





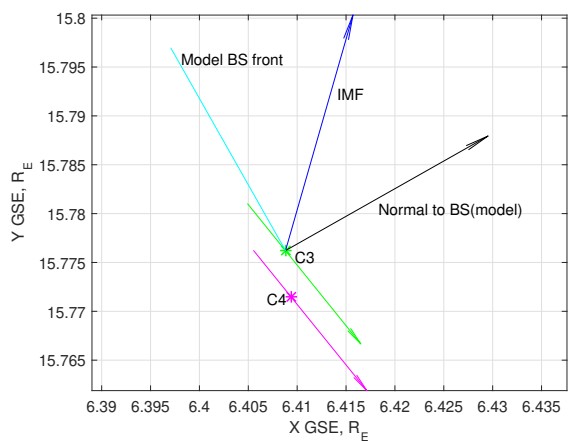

**Figure 6.** Spacecraft orbit and model shock position for shock 12 December 2011.

**Table 1.** Wave analysis data for shock 18 December 2011, 14:37:27–14:37:47.

| | |
|---|---|
| max eigenvector, $V_{max}$ | -0.23, 0.94, 0.27 |
| med eigenvector, $V_{med}$ | 0.97, 0.20, 0.15 |
| min eigenvector, $V_{min}$ | -0.08, -0.29, 0.95 |
| eigenvalues | 2.23, 6.64, 11.50 |
| magnetic field C3, $B_3$ (nT) | -3.58, 9.53, 0.96 |
| local proton velocity C4 (km/s) | -118.1, 82.1, -29.29 |
| angle, $V_{max}$ and IMF | $34^o$ |
| angle, $V_{min}$ and IMF | $110^o$ |
| angle, $V_{max}$ and $B_3$ | $12^o$ |
| angle, $V_{min}$ and $B_3$ | $99^o$ |
| peak frequency in max component | 0.3 Hz |
| time shift in magnetic field along $V_{max}$ | 0.13 s |
| separation along $V_{min}$ | 10 km |
| wavelength | 252 km |





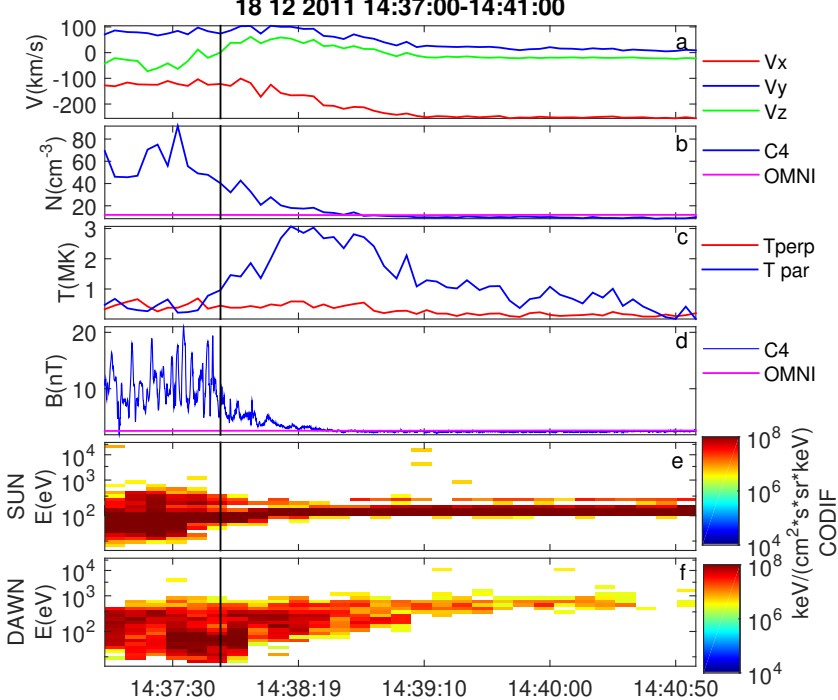

**Figure 7.** Overview of C4 magnetic and plasma measurements for event 18 December 2011. (a) proton velocity, (b) proton density and OMNI solar wind density, (c) proton parallel and perpendicular temperature, (d) magnetic field magnitude and OMNI IMF magnitude, (e,f) proton spectrograms for the sunward and dawnward looking sectors.

amplitudes during the first 20 s 16:01:15–16:01:35 UT downstream the front despite relatively small separation. This difference in amplitudes was typical for all shocks registered during this day (8 crossings within 2 hours in Table S1).

One more crossing is from January 3rd, 2008 (14:30–1435 UT) with Cluster separation ∼100 km (Table S1, Fig. S2 in Supplement). OMNI data showed very low IMF (1.1 nT) and $\beta = 39$. Local upstream magnetic field at C4 was so low only
5   episodically at 3 min before the front (Fig. S2), so very high $\beta$ can not be fully confirmed. The detailed wave activity at the front is in Fig. 12. Only relatively high frequency oscillations about 1–2 Hz are present, strongly different at two spacecraft, therefore no phase analysis is possible. There are no wave packets with the stable phase. For example, at 143410–143414 UT $X$ and $Z$ components are in anticorrelation for C3 and C4, while immediately near, at 143408–143410 UT these components are in phase. Amplitude of oscillations is comparable in components and magnitude of magnetic field. Another event from our
10   statistics with similar higher frequency variations is that of 31 December 2003 (Table S1).





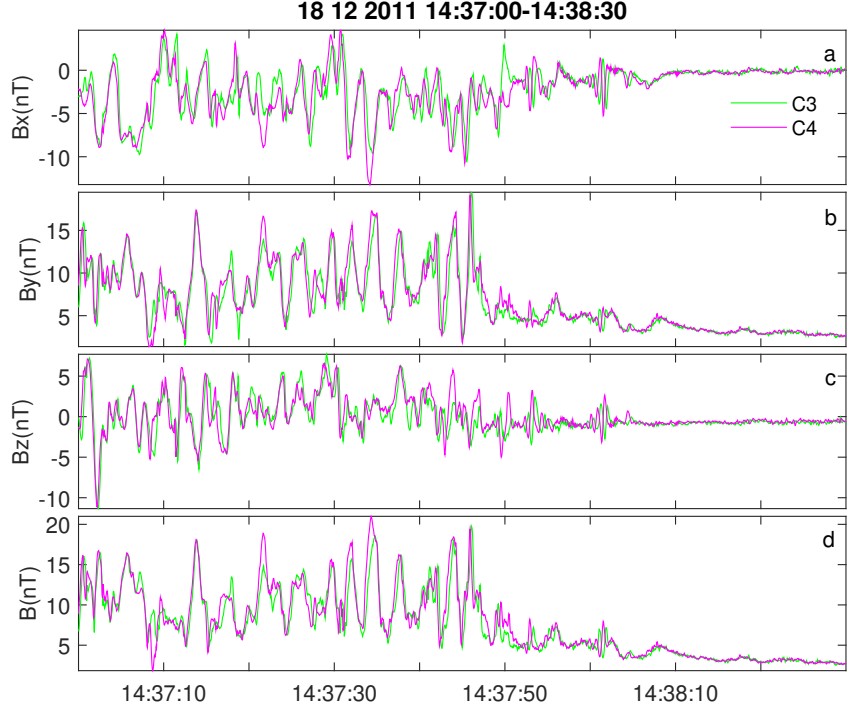

**Figure 8.** Full resolution magnetic waveform for shock 18 December 2011. In panels (a-d) are components and total value of magnetic field.

## 4 Discussion

### 4.1 Solar wind input

High-$\beta$ solar wind is relatively rare at the Earth orbit. In our study we accepted somewhat ad-hoc threshold of high $\beta$ equal to 10. Such interplanetary conditions tend to occur during solar minima, being created by slow cold dense solar wind with low

5 IMF (1–2 nT). It is not always easy to confirm that the observed shock crossing actually occurred in high-$\beta$ solar wind interval, identified in OMNI. The first set of problems is related with association of partiular shock front crossings with stable high-$\beta$ intervals. It is reasonable to consider durations of high-$\beta$ intervals at least of the order of tens of minutes, and reject candidate events at the moments of $\beta$ changes, since it is more convincing to study bow shock events under stable upstream conditions. These problems are relatively straightforward to identify and solve.

10  A more substantial problem is due to the finite spatial scales of high-$\beta$ solar wind. We measure solar wind in L1 halo orbit, 1.5 million kilometers away from Earth and with halo radius not less than 200 000 km (for ACE spacecraft). A substantial part of modern OMNI data are taken from Wind spacecraft, which is currently on a much wider halo orbit (300–400 thousand km) (Podladchikova et al., 2018). Solar wind and IMF structures encountered in L1 are not necessarily the same, that actually affect





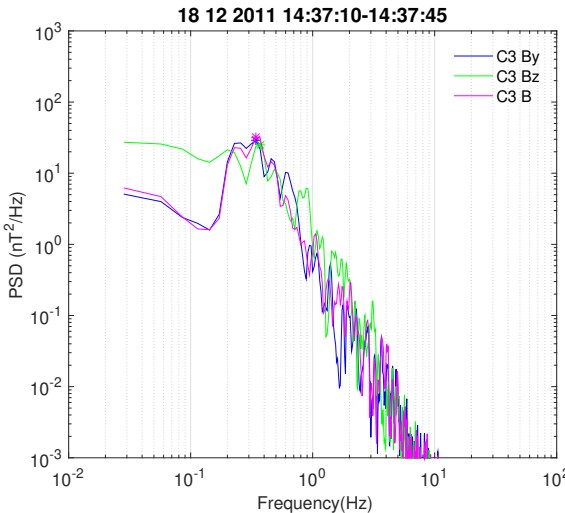

**Figure 9.** C3 frequency spectra for $B_y$, $B_z$ components and magnetic field magnitude for shock 18 December 2011.

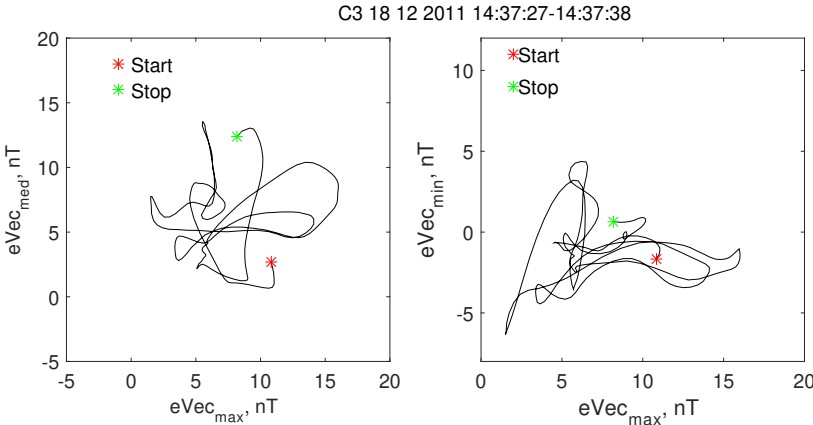

**Figure 10.** Hodographs of C3 magnetic field in eigenvector coordinates for shock 18 December 2011.

the magnetosphere. The most questionable is spatial persistence of relatively small changes of IMF from 2 to 1 nT, required for creation of very high-$\beta$ intervals.

Though the analysis of the scale of high-$\beta$ areas in solar wind was not performed with the multisatellite data, available reports on similar topics indicate significant potential problems. ISEE data suggested that during periods of medium to low



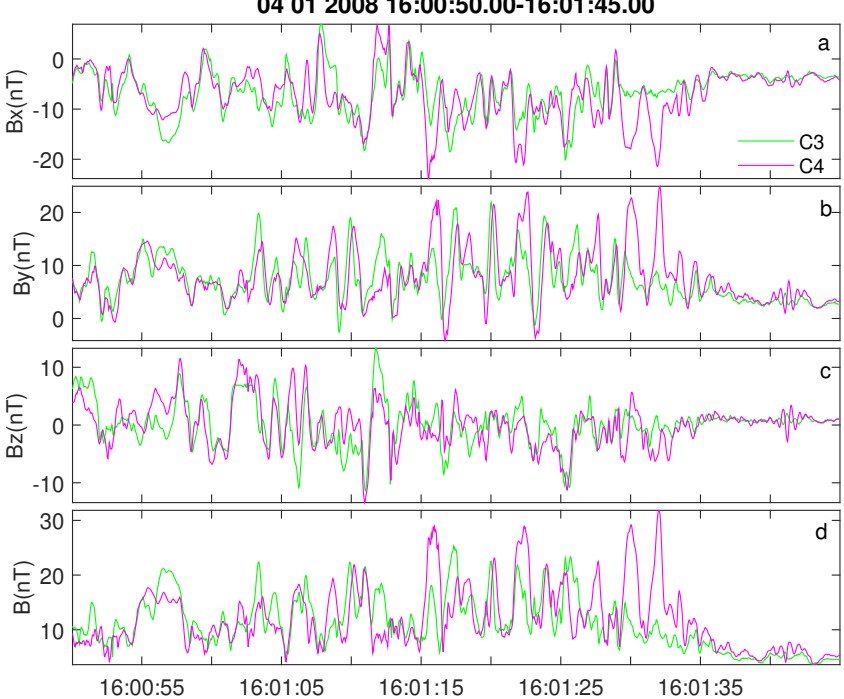

**Figure 11.** Full resolution magnetic waveform for shock 04 January 2008. In panels (a-d) are components and total value of magnetic field

variance of magnetic field, magnetic features with scale widths of 20 $R_E$ perpendicular to the IMF may occur (Crooker et al., 1982). Comparison of L1 Wind and near-Earth Interball data for 1996–1999 have shown (Petrukovich et al., 2001), that the large-scale IMF structures, associated with geomagnetic storms (with the threshold of IMF $B_z$ GSM below –10 nT during 3 hours) are practically the same in L1 and the near-Earth orbits. However, about 20–80% of smaller everyday IMF variations

5   (depending on their amplitude), causing substorms (several nT in magnitude on one-hour scale) are different by more than 25% .

    Thus applicability of very high $\beta$ values in OMNI is not automatic. It is not always possible to check solar wind $\beta$ value immediately before shock crossing with local spacecraft. A spacecraft needs to probe pristine solar wind and then rapidly cross the shock, or there should be an additional near-Earth solar wind monitor. Magnetic field can be reliably measured

10   with magnetometer (still assuming offset uncertainty of about 0.1 nT). Accuracy of solar wind density and ion temperature values is more problematic since at L1 they are measured by a specialized thoroughly calibrated instruments, while with a magnetospheric spacecraft, calibration could be rougher for the specific case of solar wind flow. Additional (relative to OMNI-based ones) very-high $\beta$ intervals may actually form near Bow shock due to local variability of solar wind and IMF. Assumptions on helium content and electron temperature, used while $\beta$ calculations, may also result in some errors.





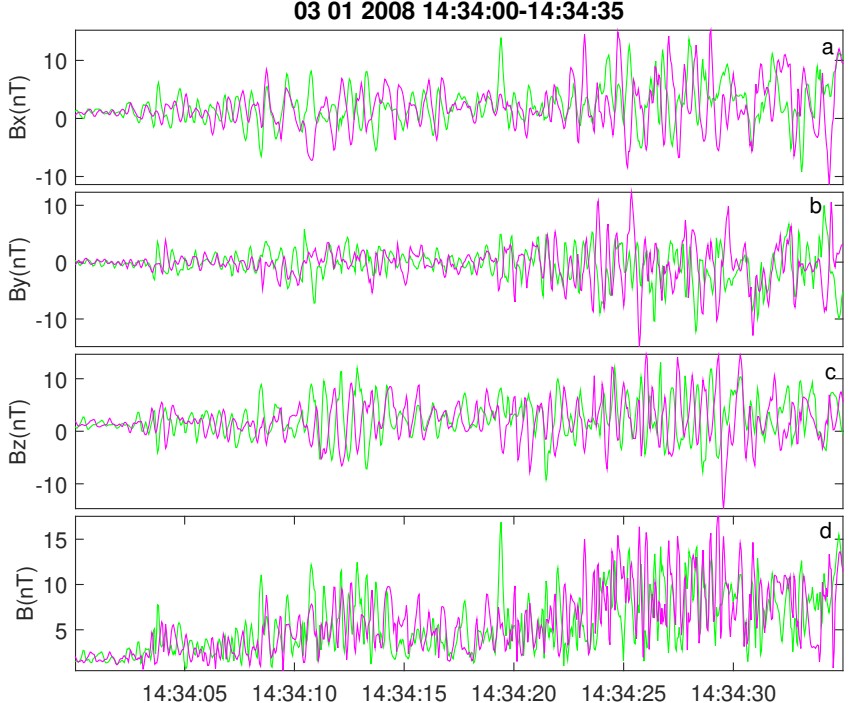

**Figure 12.** Full resolution magnetic waveform for shock for shock 03 January 2008. In panels (a-d) are components and total value of magnetic field

For the purpose of our study (properties of magnetic waves in shock front) it is also important to have in mind, that the local $\beta$ is more relevant, rather than the far upstream one. Immediately upstream the shock front magnetic field increases due to presence of rotating ions in the shock foot. In the ordinary quaziperpendicular shocks (Scudder et al., 1986) this increase is rather small. When IMF is very low (in a high-$\beta$ case), the foot increase may change (decrease) the local beta quite substantially.

5    In the zone of the strongest magnetic variations behind the shock front $\beta$ should be similar to that upstream due to Rankine-Hugoniot jump conditions.

### 4.2 Shock properties

Observed shock crossings with high-amplitude magnetic variations are generally consistent with the earlier reports for high-$\beta$ shocks. All selected shocks are quazi-perpendicular and have high alfvenic and magnetosonic Mach numbers. The observed

10   structure of high-$\beta$ shocks is quite different from that for low-$\beta$ supercritical quasi-perpendicular events (e.g. Krasnoselskikh et al., 2013). We identify shock front with the outburst of wave activity, but significant increase of ion density and magnetic field occurs also upstream of it. It was difficult to determine location of shock ramp (main increase of magnetic field magnitude,



typical for supercritical quazi-perpendicular shock), since the front actually is built with large-amplitude spikes. A study of ion kinetics, which may help in identification of detailed shock front structure, is left for future publications.

The high-$\beta$ shock front is formed by strong magnetic variations with amplitude of $\sim$20 nT at frequencies 0.1–0.5 Hz, Analysis shows irregular wave structure with no stable phase. polarization is close to linear, so that there are substantial variations in magnitude of magnetic field also. In several shock examples it was possible to determine spatial scale of these variations around 200–900 km. Doppler shift determination is not reliable enough, since wave vector direction is not known for linear polarization. On this background much lower amplitude (1–2 nT) variation are often present with 1–2 Hz frequency. In two events wave activity is dominated by 1–2 Hz variations, also with irregular phase. In some events strong differences between two Cluster spacecraft suggested spatial scales of the order of tens km.

Properties of magnetic variations suggest their compressional nature (linear polarization with strong changes in total magnetic field) and strong spatial localization due to absence of stable wave packets. Thus variations are strongly different from that in low-$\beta$ events where clear whistler wave packets with elliptic polarization dominate. Observed polarization is also not consistent with earlier suggested alfven mode (Kennel and Sagdeev, 1967a). Pokhotelov and Balikhin (2012) suggested a Weibel mode in the finite magnetic field, developing a mix of two opposite circular polarizations. Reliable wavelength determination would be a key to final wavemode identification.

## 5   Conclusions

High-$\beta$ ($\beta > 10$) shocks are relatively rare and largely unexplored class of Earth bow shock. Formation of high-$\beta$ interplanetary plasmas is mostly related with dense slow solar wind and very low magnetic field up to 1–2 nT. The higher is $\beta$ (in OMNI), it is more difficult to confirm it locally.

Our shock analysis was limited to quazi-perpendicular cases and shows substantial differences from similar supercritical shocks with lower $\beta$. Dominating magnetic waves show more irregular linear polarization. An extended analysis with Magnetospheric Multiscale mission data including electron and ions is necessary to conclude the wave mode analysis.

*Author contributions.* OMC and PIS performed the data processing and analysis. AAP is responsible for data analysis and interpretation. AAP prepared the manuscript with contributions from all co-authors.

*Competing interests.* The authors declare that they have no conflict of interest.

*Acknowledgements.* The data analysis was funded with Russian Science Fund project 05-14-00824. We are thankful for Cluster Science Archive, CDAWeb and OMNI for availability of spacecraft data.



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
