# Peer review of "Wave activity in front of high- $\beta$ Earth bow shocks"

_Annales Geophysicae, 2018_

## Referee Comment (RC1) · Anonymous Referee #1 · 28 Oct 2018

General comments

The main goal of the paper is to study the properties of the Earth's bow shock during intervals of high plasma beta (>10). The authors initially employed a semi-automated search routine to identify intervals of high beta when spacecraft were within 5 Earth radii of the model bow shock. In the end, 22 suitable bow shock crossings were identified for study. After reading the manuscript, I am unable to recommend the paper for publication due to many issues which I will present in detail below. I do feel that the reviewers have the concept and the data to make an interesting study, however the current form is too poorly executed with numerous issues and needs significantly more analysis. Since the paper does not reach any meaningful or clear conclusion, then I the current iteration cannot justify publication. It is my recommendation that the authors

extensively improve the paper and re-submit when it has been sufficiently revised.

Specific comments

The title suggests that waves upstream of the bow shock are studied, however in each case, waves downstream, or in the shock transition are investigated.

After reading it, I am unable to identify any clear conclusion or result from the analysis. Several shock crossings are presented, and in each one waves are shown. The authors: 1) do not identify the wave-mode, 2) determine the role of the waves in dictating the shock structure, 3) determine what the relationship between high beta and the waves are, 4) compare with shocks of lower beta, 5) consider the role of the waves in dissipating energy at the shock front.

The authors claim on line P7 L4-5 that "the observed front structure is very different in comparison with that expected for a supercritical shock". The shock profile appears to me to be very similar to a quasi-parallel shock and nothing remarkable. Since the geometry is 46degrees based on a model, then based on the error of a model shock normal, the quasi-parallel/perpendicular geometry cannot be confirmed. However, based on the profile, it appears to be a quasi-parallel shock structure, and thus a prolonged and turbulent upstream-downstream transition is expected.

The authors identified 22 suitable bow shock crossings, however these 22 only occur on 7 days. For example, 12/22 take place on 3-4/01/2008. This should me mentioned in the paper.

In the conclusions, there are many issues, for example: 1. It's known what caused high beta solar wind, one would only have to perform some rudimentary analysis of the OMNI data. This is not necessarily a result. 2. It is stated "Our shock analysis was limited to quazi-perpendicular cases", this is not true, many of the shocks are around the 45degree threshold and look like quasi-parallel shocks (even more so considering possible error of the model normal). Some shocks in the supplementary table are

<45degrees. The authors seem to only consider the quasi-perpendicular shock geometry, they need to also include quasi-parallel. 3. It is said that differences are observed compared to lower beta shocks, but this is barely discussed and verified. The authors should select shocks with similar geometry but with lower plasma beta to compare. In fact, on the first line of the discussion it's said that their observations are similar to already reported structure of high-beta shocks. So, it's unclear what the new results are from this study.

Why are these particular shocks selected for study and what is their significance? There are other shocks which are more clearly defined as quasi-perpendicular in the supplementary table. For example, the 2 Jan 2008 has a geometry of 83degrees. This would likely have a more clearly defined ramp structure and would be easier to determine the shock properties. Please justify the event selection.

I find it unfortunate that 22 shocks are identified but no effort is made to use all these data. How do the few selected shocks compare to the other ones identified? Are the shock profiles and structure comparable? Are similar waves seen for all of them? Are the waves associated with the geometry or Mach number?

The geometry of the shocks found is from 39degrees to 84degrees. Thus, these shocks will have very different structure as they cover the quasi-parallel to quasi-perpendicular regimes. What effects reported are from the high-beta upstream condition, and what are simply from the geometry.

In the final sentence it is said MMS is needed to conclude the wave-mode analysis, why not Cluster; this has been done before. The main issue with wave-modes is to separate temporal and spatial variations. On larger scales (larger than ramp width) then Cluster is a better dataset. Is there no shocks in which the wave-telescope or phase-shift analysis can be performed? Also, if MMS is required then why not use MMS? Maybe there were not enough events?

P7 L18: I am unsure how reliable is the calculation of the wavelength is based on a

delay is so small at 0.13seconds, the waveforms look almost instantaneous? I am not convinced that one can separate temporal and spatial variations on this scale for this spacecraft separation. Also, what is the angle between the spacecraft separation and the direction which was used to determine the delay. If it is perpendicular, then this would also significantly increase the error.

Hodograms: it is difficult to get any useful information from Figure 10. In fact, there is no clear polarisation. Did the authors try to compute this over one or two wave cycles? This might be more meaningful. The min-int hodogram should also be plotted.

More detailed explanation should be given to the observations. Can the authors discuss the parallel heating upstream of the shock in Figure 7.

In summary: the paper requires too much work and revision for revisions. To give the authors enough time, then I suggest the paper is rejected this time but I would encourage them to re-submit when it has been sufficiently revised.

---

## Referee Comment (RC2) · Anonymous Referee #2 · 30 Oct 2018

**Review on the paper angeo-2018-110**

The paper is devoted to analysis of high-$\beta$ shocks. The subject is timely, there are no many studies of such shocks. The problem of the paper that analysis has not been done. The manuscript leaves the impression that the study has only begun. There paper does not arrive to any conclusions. There is attempt to convert temporal measurements to spatial one and, therefore, relate the nice figures to shock physics (eg ion motion). It is not clear what is called "waves". The choice of a model normal against other methods is not justified. It is now even said what is "full resolution" of the magnetic field measurements. Are readers supposed to know that ? Except statistics of high-$\beta$ occurrence and nice magnetic profiles (the third one seems to be a typical quasi-parallel shock), there is no new physics in the paper. It should be returned to the authors for completion of the study before submitting again.

---

## Author Comment (AC1) · 10 Jan 2019

**Reply to Referee #1.**

General comments

*The main goal of the paper is to study the properties of the Earth's bow shock during intervals of high plasma beta (>10). The authors initially employed a semi-automated search routine to identify intervals of high beta when spacecraft were within 5 Earth radii of the model bow shock. In the end, 22 suitable bow shock crossings were identified for study. After reading the manuscript, I am unable to recommend the paper for publication due to many issues which I will present in detail below. I do feel that the reviewers have the concept and the data to make an interesting study, however the current form is too poorly executed with numerous issues and needs significantly more analysis. Since the paper does not reach any meaningful or clear conclusion, then I the current iteration cannot justify publication. It is my recommendation that the authors extensively improve the paper and re-submit when it has been sufficiently revised.*

We are grateful to the reviewer for the attentive reading and detailed comments and advices. But we cannot agree, that the paper does not reach any conclusion. We determine the time scales of the shock transition, frequency and polarization for the highest-amplitude magnetic variations and estimate their spatial scale. There is no definite conclusion on the wave mode, but this is because linearly polarized waves not allow to determine reliably the wave vector direction. Besides that, there was a large amount of hidden work for initial data selection, since these are relatively rare shocks, almost never described before. We selected all crossings, suitable for multipoint analysis, at least in principle (out of almost 20 years of Cluster observations). Yes, about other shock types a lot more is known and results are more quantitative. However, this was achieved after tens of publications and thousands of crossings studied. It is not unexpected to achieve not so high level of details in this very first publication using very rare events.

The paper was initially written in a very concise manner to avoid discussion of secondary phenomenological features. However, this may result in a somewhat misleading presentation. Now the description is substantially extended. We added more details on the sample crossings, review generality of results for the whole statistics and discuss comparison with other publications. More specific replies are below.

**Corrections in the text:** the manuscript is extended by almost 50% and improved. In Sec 1 and 2 changes are marked by bold. Sections 3 and 4 (data examples and discussion) are rewritten almost completely. The full analysis of Event #2 (similar to that for Event #1) is given. Description on Event #3 is significantly extended, including a new note on substantial variability of background magnetic field. Summary on statistics of events is added. In total 3 figures and 1 table are added in the main text and 2 figures in Supplement.

*The title suggests that waves upstream of the bow shock are studied, however in each case, waves downstream, or in the shock transition are investigated.*

Ok. We assumed the term "front" in a very general sense. We study the dominating high-amplitude variations immediately downstream, since their amplitudes are larger than background field, these variations are actually a part of the transition. **Now we corrected the title.**

*After reading it, I am unable to identify any clear conclusion or result from the analysis. Several shock crossings are presented, and in each one waves are shown. The authors: 1) do not identify the wave-mode, 2) determine the role of the waves in dictating the shock structure, 3) determine what the relationship between high beta and the waves are, 4) compare with shocks of lower beta, 5) consider the role of the waves in dissipating energy at the shock front.*

As written above, we conclude on frequency, polarization, estimate of spatial scale, general structure of shock transition. Without definition of wavelength and wave vector directions it is impossible to conclude on the wave mode and on the contribution to dissipation and structure. We specially collected all available Cluster project statistics of multipoint crossings. However, it proved to be impossible even with these data to determine reliably wave propagation direction (because this are high-amplitude linearly polarized wave, may be actually – growing localized magnetic clumps).
**Text added:** more explanation in the text and comparison with the previous results on low beta shocks (more details are below in a reply to the similar comment).

*The authors claim on line P7 L4-5 that "the observed front structure is very different in comparison with that expected for a supercritical shock". The shock profile appears to me to be very similar to a quasi-parallel shock and nothing remarkable. Since the geometry is 46degrees based on a model, then based on the error of a model shock normal, the quasi-parallel/perpendicular geometry cannot be confirmed. However, based on the profile, it appears to be a quasi-parallel shock structure, and thus a prolonged and turbulent upstream-downstream transition is expected.*

We now add information on shock angle using other methods. All variants are not ideal. Coplanarity definition has its own errors. In two presented cases the shock normals are almost the same with all variants. In Event #3 with very low magnetic field, its direction varies and coplanarity approach cannot be applied. We also discuss some possible implications.
The transition lasts couple of minutes and has very laminar gradual change in ion moments, thus it is similar on a large scale to other oblique or quasi-perpendicular shocks. However, on a smaller scale the structure is somewhat different since there is no clear localized magnetic jump. The main increase of magnetic field is smeared and dominated by variations in all events."Typical" quasi-parallel shock (e.g. Burgess et al. and earlier references therein) has a prolonged transition up to several Earth radii long with patchy ion interaction with individual intensifications (SLAMS). Thus our events are more similar to reforming oblique shock (e.g. Lefebvre et al 2009) or quasi-perpendicular shock in the decay phase.
**ext added:** Results on theta-Bn with other methods. More careful definition of events as "oblique" shocks. A more detailed discussion of comparison with the quasi-parallel shock.

*The authors identified 22 suitable bow shock crossings, however these 22 only occur on 7 days. For example, 12/22 take place on 3-4/01/2008. This should me mentioned in the paper.*

OK. **Mentioned.** This issue does not affect validity of conclusions. In fact, it is even highlights stability of the observed profile for Event 3 (the nearest neighbor in 10 min has no "high-frequency" pattern).

*In the conclusions, there are many issues, for example: 1. It's known what caused high beta solar wind, one would only have to perform some rudimentary analysis of the OMNI data. This is not necessarily a result.*

This in fact is not an independent result on origins of high-beta in solar wind. It is more or less clear, even intuitively, that such conditions correspond to high density and low magnetic field. The main purpose of this study is to determine implications for high-beta shocks: how often such events can be found, how transient is magnetic field. This is **very important methodical issue**, since high-beta intervals correspond to extremely small and hence variable magnetic field. Such magnetic input might have important consequence for shock structure. The decreasing role of magnetic field in higher beta shocks is actually one of important motives for this study.

**Text added:** More explanations on the use of our solar winds statistics result in the end of Sec 2 and in discussion

*2. It is stated "Our shock analysis was limited to quazi-perpendicular cases", this is not true, many of the shocks are around the 45degree threshold and look like quasi-parallel shocks (even more so consider-ing possible error of the model normal). Some shocks in the supplementary table are <45degrees. The authors seem to only consider the quasi-perpendicular shock geom- etry, they need to also include quasi-parallel.*

Actually in the literature, there are two approaches. With one quasi-perpendicular and quasi-parallel shocks are separated by 45 degrees. With the other, "the middle" around 45 degrees is attributed to oblique shocks. Quasi-parallel shocks are then that with angles closer to zero. In fact, the second approach is more correct. We now more carefully call our shocks as oblique and quasi-perpendicular. Oblique and quasi-perpendicular shocks are, in a sense, similar, having a well-defined transition (on a scale of minutes). All our events well fit this sort. Quasi-parallel shocks are quite different having very prolonged and patchy transition on the scale of Earth radii. Some discussion of this issue is included.

**Text added:** More careful definition of events as "oblique" shocks. A more detailed discussion of comparison with the quasi-parallel shock.

*3 It is said that differences are observed compared to lower beta shocks, but this is barely discussed and verified. The authors should select shocks with similar geometry but with lower plasma beta to compare.*

There are thousands of crossings of oblique shocks, and many investigations are published with very diverse results. It would be difficult to justify the prompt selection of those for comparison. Some initial comparison was done in Winterhalter et al 1988 (see the next reply also). Thus we concentrated in this first study on common features of high-beta shocks. This is an essential step,

since it is important first to understand what are the primary properties to be compared. We used previous publications on oblique shocks for comparison.

**Text added:** Some results on waves in lower beta shocks are now cited and compared in Discussion.

*In fact, on the first line of the discussion it's said that their observations are similar to already reported structure of high-beta shocks. So, it's unclear what the new results are from this study.*

The detailed connection of this report with previous publications is provided in the Introduction. Earlier only very general conclusions on "very high amplitude" of variations were available in comparison with the lower-beta variants (Winterhalter et al) The first phrase in Discussion only means that our study does not contradict this previous result. In this report we determine frequency, polarization and estimate the spatial scale of these variations, as well as make some conclusions on general structure of these shocks. If this phrase is so disorienting, we reformulate it in more detail.

**Text added:** A more detailed statement in Sec 3.4 and Sec.4

*Why are these particular shocks selected for study and what is their significance? There are other shocks which are more clearly defined as quasi-perpendicular in the supplementary table. For example, the 2 Jan 2008 has a geometry of 83degrees. This would likely have a more clearly defined ramp structure and would be easier to determine the shock properties. Please justify the event selection.*

Three typical events were taken. The only full justification can be to plot all 22 events. As for the particular event with 83 degrees – it is actually not different, there is no clear ramp, magnetic field grows gradually (see the Figure below). The explanation can be, that for such low magnetic fields, their exact value and orientation might be not so important for shock properties.

[Figure]

*Overview of shock from 2008-Jan-2*

*I find it unfortunate that 22 shocks are identified but no effort is made to use all these data. How do the few selected shocks compare to the other ones identified? Are the shock profiles and structure comparable? Are similar waves seen for all of them? Are the waves associated with the geometry or Mach number?*
*The geometry of the shocks found is from 39degrees to 84degrees. Thus, these shocks will have very different structure as they cover the quasi-parallel to quasi-perpendicular regimes. What effects reported are from the high-beta upstream condition, and what are simply from the geometry.*

Actually there is only one event with 83 degrees and one with 64 degrees. All other angles are around 45. There is no "truly" quasi-parallel shocks with the angles close to zero. All shocks have similar large Ms number. Thus it would be misleading to present any statistics (of transition length or frequency of variations) depending on the angle or the Mach number.
**Text added:** We now included a more detailed reference to statistics, pointing similarity of all shocks with the presented examples.

*In the final sentence it is said MMS is needed to conclude the wave-mode analysis, why not Cluster; this has been done before. The main issue with wave-modes is to separate temporal and spatial variations. On larger scales (larger than ramp width) then Cluster is a better dataset. Is there no*

*shocks in which the wave-telescope or phase-shift analysis can be performed? Also, if MMS is required then why not use MMS? Maybe there were not enough events?*

The full reasoning of data set selection is included in Sec. 2. Out of about a 100 of candidate events (from all projects) we selected 22 Cluster events with separations about couple hundred km and smaller right in the attempt to perform multipoint analysis of ramp structures around 0.1-0.5 Hz (actually all our studied variations can be in a sense attributed to the smeared ramp structure). Wave telescope can be used with four-point data, but we have only one event from 2003, when the regular tetrahedron was available. Unfortunately, this event has waves "type-2" and the separation of 300 km was too large to determine the scale. All other events are from the later years of Cluster, when only two spacecraft C3 and C4 were close sometimes. Such poor statistics is just because high-beta solar wind is rare. With the two-point data the spectral or phase shift analysis can provide the scale only if wave propagation direction is known. Exactly such analysis is performed in this study. We choose the time-domain analysis, since the main frequency of oscillations is well defined. To perform the wave-telescope analysis with no supposition on wave propagation we need (potentially!) the four-point MMS data with the regular separation about 10 km. Since MMS is new project, there is no so much statistics and one needs to wait. This analysis will be a part of future publications.

**Text added:** Our discussion is extended to make this point clearer.

*P7 L18: I am unsure how reliable is the calculation of the wavelength is based on a C3 delay is so small at 0.13seconds, the waveforms look almost instantaneous? I am not convinced that one can separate temporal and spatial variations on this scale for this spacecraft separation.*

In our case 0.13 sec corresponds to 2-3 sampling intervals. It is quite sufficient to perform reliable phase shift analysis. In fact, the stable time shift of two curves is well observed even visually in Fig. 8.

**Text added:** information on validity of phase shift is added.

*Also, what is the angle between the spacecraft separation and the direction which was used to determine the delay. If it is perpendicular, then this would also significantly increase the error.*

Table 1 includes also spacecraft distance along the min-var direction (10 km, compared with 30 km full distance). The range of possible scales for different propagation direction is given in the text. Of course, the wavelength can be very small in a rather unlikely case, when the spacecraft are exactly perpendicular to the wave vector.

**Text added:** Actually this information on separation is present in the text.

*Hodograms: it is difficult to get any useful information from Figure 10. In fact, there is no clear polarisation. Did the authors try to compute this over one or two wave cycles? This might be more meaningful. The min-int hodogram should also be plotted.*

All full hodograms are now included. Indeed the main conclusion is that there is no stable polarization. The given time interval was selected because of the stable phase shift between the spacecraft. Selection of even shorter one with two "nice" cycles will not result in any changes in determination of the spatial scale, because (1) phase shift is the same, (2) max var direction (important to fix propagation direction in linear polarization case) is the same.

**Text added:** Additional hodograms are included + more explanations.

*More detailed explanation should be given to the observations. Can the authors discuss the parallel heating upstream of the shock in Figure 7.*

A substantially longer description is now included. The increase of parallel temperature looks natural since ions tend to escape upstream along the magnetic field, so the spread of their velocity is converted in temperature. The detailed analysis of ion dynamics is left to future publications.

*In summary: the paper requires too much work and revision for revisions. To give the authors enough time, then I suggest the paper is rejected this time but I would encourage them to re-submit when it has been sufficiently revised.*

The manuscript is significantly rewritten. Observation and discussion section is changed almost completely.

---

## Author Comment (AC2) · 10 Jan 2019

Reply to Referee #2 The paper is devoted to analysis of high-$\beta$ shocks. The subject is timely, there are no many studies of such shocks. The problem of the paper that analysis has not been done. The manuscript leaves the impression that the study has only begun. There paper does not arrive to any conclusions. There is attempt to convert temporal measurements to spatial one and, therefore, relate the nice figures to shock physics (eg ion motion).

We are grateful to the reviewer for the attentive reading. We cannot agree, that the paper does not reach any conclusion. We determine the time scales of the shock transition, dominating frequency for the highest-amplitude magnetic variations, their

polarization and estimate the spatial scale. There is no definite conclusion on the wave mode, but this is because linearly polarized waves not allow to determine reliably the wave vector direction. Besides that, there was a large amount of hidden work for initial data selection, since these are relatively rare shocks, almost never described before. We selected all crossings, suitable for multipoint analysis, at least in principle (out of almost 20 years of Cluster observations). Yes, about other shock types a lot more is known and results are more quantitative. However, this was achieved after tens of publications and thousands of crossings studied. It is not unexpected to achieve not so high level of details in this very first publication using very rare events. The paper was initially written in a very concise manner to avoid discussion of secondary phenomenological features. However, this may result in a somewhat misleading presentation. Now the description is substantially extended. We added more details on the sample crossings, review generality of results for the whole statistics and discuss comparison with other publications. More specific replies are below.

Corrections in the text: the manuscript is extended by almost 50% and improved. In Sec 1 and 2 changes are marked by bold. Sections 3 and 4 (data examples and discussion) are rewritten almost completely. The full analysis of Event #2 (similar to that for Event #1) is given. Description on Event #3 is significantly extended, including a new note on substantial variability of background magnetic field. Summary on statistics of events is added. In total 3 figures and 1 table are added in the main text and 2 figures in Supplement.

It is not clear what is called "waves".

The terms are corrected and streamlined in the whole text. Now we use "magnetic variations" as more neutral term. "Waves" indeed assume some repeating spatial structure.

The choice of a model normal against other methods is not justified.

We add comparison with other methods of normal calculation. They actually provide

the same results in Events #1 and #2. In Event #3 with very low magnetic field, its direction varies and coplanarity approach cannot be applied. We also discuss some possible implications.

It is now even said what is "full resolution" of the magnetic field measurements. Are readers supposed to know that?

"Full resolution" was adopted from the name of the data set in Cluster Science archive. For these events it is about 20 Hz sampling (now mentioned in the text).

Except statistics of high-$\beta$ occurrence and nice magnetic profiles (the third one seems to be a typical quasi-parallel shock), there is no new physics in the paper.

We do not agree. We determine also frequency, polarization of dominating high-amplitude variations and estimate it spatial scale. Some methods to calculate wave propagation direction are also tested, but proved not applicable. We also make initial suppositions about wave mode. (More detailed reply is above in the beginning)

(the third one seems to be a typical quasi-parallel shock)

Though there is no clear ramp (jump of magnetic field) in Event 3, the transition is quite localized in several tens of seconds and has very laminar gradual change in ion moments. "Typical" quasi-parallel shock (e.g. Burgess et al. and earlier references therein) has a prolonged transition up to several Earth radii long with patchy ion dynamics. The detected profile is very similar in time scales for all studied crossings and could be also observed for reforming oblique (Lefebvre et al 2009) or supercritical quasi-perpendicular shocks. Text added: This issue is now also discussed in more detail.

finally, the new revised paper is in supplement for reference

Please also note the supplement to this comment:
https://www.ann-geophys-discuss.net/angeo-2018-110/angeo-2018-110-AC2-

supplement.pdf

[Figure]

**Supplement:**

**Low frequency magnetic variations at high-$\beta$ Earth bow shocks**

Anatoli A. Petrukovich[1], Olga M. Chugunova[1], and Pavel I. Shustov[1]

[1]Space Research Institute of Russian Academy of Sciences, Moscow, Russia

**Correspondence:** A.A.Petrukovich (a.petrukovich@cosmos.ru)

**Abstract.** Earth's bow shock in high $\beta$ (ratio of thermal to magnetic pressure) solar wind environment is a rare phenomenon. However such an object is ubiquitous in astrophysical plasmas. Typical solar wind parameters related with high $\beta$ (here $\beta > 10$) are: low speed, high density and very low IMF 1–2 nT. These conditions are usually quite transient and need to be verified immediately upstream of the observed shock crossings. We survey statistics of high-$\beta$ shock observations by near-Earth spacecraft since 1995. About a hundred crossings were initially identified mostly with oblique or quasi-perpendicular geometry and high Mach number. In this report 22 crossings by Cluster project are studied with multipoint analysis, allowing to determine spatial scales. Observed shock structure is different from that for supercritical shocks with $\beta \sim 1$. The main magnetic field increase is smeared to couple tens of seconds and is dominated with magnetic variations $\sim$0.1–0.5 Hz (in some events — 1–2 Hz). Their polarization has no stable phase and is closer to linear, while spatial scales are of the order of hundred km at 0.1–0.5 Hz.

*Copyright statement.* TEXT

**1 Introduction**

Shocks are the primary dissipation mechanism in space plasmas with supersonic flows (Sagdeev, 1966; Kennel et al., 1985; Krasnoselskikh et al., 2013). A brand new branch of plasma science, theory of collisionless shocks, appeared in the sixties, in response to new space observations. In the solar system solar wind forms the bow shocks at planets and comets, as well as the termination shock at the heliospheric interface. **Interplanetary shocks develop inside the heliosphere after solar eruptions, when large-scale transient structures propagate relative to the regular solar wind flow.** In the distant space, shocks are associated with supernova explosions, stellar winds, collisions of galaxy clusters and are believed to have a leading role in the acceleration process of cosmic rays (Axford et al., 1977; Krymskii, 1977). Physics of space shocks was reviewed in AGU Geophysical Monographs, volumes 34 and 35 (1985). The Earth bow shock has been most thoroughly studied since the launch of the first spacecraft and is the main source of our in-situ knowledge of collisionless shock structure and dynamics.

Electromagnetic fields and waves in collision less plasma shocks are of primary importance. Due to presence of magnetic field a wide variety of shock types exists with quite differing structure (Kennel et al., 1985). **Magnetic field vector enters Rankine-Hugoniout equations, defining the relation between upstream and downstream conditions.** In the absence of

collisions, kinetic mechanisms of field-particle interactions are responsible for dissipation and particle acceleration (Sagdeev, 1966; Krasnoselskikh et al., 2013). **With quasi-perpendicular shock geometry (when the angle between the shock normal and the upstream magnetic field vector is closer to $90^o$ ) ions cannot escape upstream and relatively sharp shock transition forms with the ion-scale width (several thousand km). With quasi-parallel geometry (the angle is closer to $0^o$) ions**
5  **easily escape upstream along the magnetic field and the shock transition smears to the scales around several Earth radii (Scudder et al., 1986; Burgess et al., 2005). Oblique shocks (angles around $45^o$) are in a sense intermediate in properties, when ions partially are capable to escape upstream, but generally have rather spatially localized transition similar to quasi-perpendicular ones.**

    **Besides these large-scale magnetic field structure** also of interest at the Earth's bow shock are relatively low frequency
10  magnetic variations (from one tenth to few Hz) with visually maximal amplitudes, which actually form the primary shock front structure, dissipating ions. For example, in a supercritical quasi-perpendicular shock, the oblique whistler waves near the lower-hybrid frequency ($\sim$5 Hz) form the magnetic ramp via the non-linear steepening and decay cycle (Krasnoselskikh et al., 2002, and references therein). In the several studies the wavelength of these waves and the scale of shock ramp were determined to be around 10-s of km and oscillations were in fact identified as whistlers (Petrukovich et al., 1998; Walker et al., 2004;
15  Hobara et al., 2010; Schwartz et al., 2011; Dimmock et al., 2013; Krasnoselskikh et al., 2013). **Cyclic shock reformation is typical also for quasi-parallel shocks with substructures known as SLAMS and oblique shocks (Lefebvre et al., 2009). The specifics of a plasma wave mode, driving the front reformation depends of local plasma parameters, Mach number, etc. Immediately downstream of the shock front plasma waves at the frequencies below the ion cyclotron one were attributed to mirror, ion cyclotron, intermediate modes (e.g., Balikhin et al., 1997; Czaykowska et al., 2001).** Yet one
20  more issue of interest is electron heating. It requires sufficiently small scale variations, for non-adiabatic acceleration and following isotropisation (Balikhin et al., 1993; Vasko et al., 2018).

    Of interest to several astrophysical applications are shocks in a weak magnetic field environment (high-$\beta$ shocks), **common in interstellar and intergalaxy space** (e.g., Markevitch and Vikhlinin, 2007; Donnert et al., 2018). $\beta$ is a dimensionless parameter, a ratio of plasma thermal to magnetic energy density. Unfortunately, observations of high $\beta$ shocks near Earth
25  are quite rare, since the solar wind plasma usually has $\beta \sim$1. Very few such investigations were published, merely checking validity of Rankine-Hugoniot conditions and marking very high amplitude of magnetic variations (Formisano et al., 1975; Winterhalter and Kivelson, 1988; Farris et al., 1992). Note, that is some investigations moderate $\beta \geq 1$ was termed as "high-$\beta$" regime (e.g., $\beta = 2.4$ in Scudder et al., 1986).

    **We perform an extended experimental study of high-$\beta$ bow shock with a goal to better understand essential plasma**
30  **processes and instabilities driving shock front formation and plasma dissipation. We scanned 1995–2017 observations by Interball (1995–2000), Geotail (since 1995), Cluster (since 2000) and THEMIS (since 2007). We use $\beta >$10 criterion, which is justified in the course of this presentation. In this initial investigation we present a first, to the best of our knowledge, quantitative multi-point analysis of dominating low-frequency magnetic varations at high-$\beta$ shock transition using observations of Cluster project. We also present the occurrence of high-$\beta$ solar wind. Though such solar wind**

**statistics is generally known (review in Wilson et al., 2018), some issues relevant to shock search and analysis are still worth addressing.**

For initial selection we use orbital and spin-averaged magnetic field data from CDAWeb archive. For the detailed analysis we used full-resolution Cluster FGM magnetic field **(here with the sampling ∼20 Hz)** (Balogh et al., 2001) and HIA/CODIF ion data **(sampling once in 8 seconds)** (Rème et al., 2001) from Cluster Science Archive. Solar wind and IMF data were taken from OMNI-2 data set, the 1-hour variant was used for the initial survey, and the 1-min variant — for the final categorization of crossings. $\beta$ values are precalculated in OMNI-2, assuming constant electron temperature, He++ fraction and temperature. To access possible solar wind variability we use also ACE and Wind final Earth-shifted data from OMNI archive. All vectors are in GSE frame of reference.

**2    Solar wind statistics and details of search procedure**

We use 1-hour OMNI data for the period 1995–2017 to determine the occurrence of high $\beta$ solar wind for the subsequent shock analysis. The average solar wind $\beta$ is somewhat large than unity. High $\beta$ conditions are unevenly distributed across solar cycles (Fig. 1), being more frequent at the solar minima 1996–1997 and 2007–2009. For the threshold $\beta > 10$ there are 50–500 hours per year, while for $\beta > 20$, the number is about 3–5 times smaller.

Figure 2 shows distributions of magnetic field magnitude, solar wind speed, density and total static pressure for the full dataset of one-hour values during 1995–2017 and for the subset $\beta > 10$. The high $\beta$ corresponds to slow, cold and dense solar wind with low magnetic field (ion temperature not shown here). However the total static (magnetic plus thermal) pressure distribution is similar (Fig. 2b). Thus the high-$\beta$ events are mostly depressions of magnetic field, compensated (at least on average) by increase of plasma density. The only notable difference of distributions for $\beta > 20$ (Fig. 2a, red line) is more frequent presence of magnetic field ∼1 nT, with the average 1.6 nT, while for $\beta > 10$ the average is ∼2.2 nT.

More than 50% of events with $\beta > 10$ have one-hour duration (one point in the analyzed OMNI variant, not shown here). A sample event is in Fig. 3. There is about one-hour long decrease of magnetic field and density increase, corresponding to $\beta \sim 20$. At an occasional depletion of magnetic field below 2 nT $\beta$ jumps to about 40–80 for few minutes. Since formation of high $\beta$ conditions mostly depends on subtle variations of magnetic field magnitude around 1–2 nT (note, that $\beta$ has square dependence on magnetic field), it should be quite sensitive to spatial inhomogeneity of solar wind and IMF, and, in particular, to differences between those detected at L1 (in OMNI dataset) and actually hitting Earth. Fig. 5 shows comparison of $\beta$ calculation for Wind and ACE 1-hour data (only for times, when Wind data were used in OMNI). The scatter is indeed large.

**We formulate several conclusions important for our specific shock analysis. (1). Solar wind intervals with high $\beta =$10–20 are rare, but not extremely rare, and occur mostly during solar minimum. Thus some spacecraft (or the project phases with the specific orbit or spacecraft separation) may almost completely miss such events. (2). Duration of intervals of interest is relatively short, thus selection of shocks with stable upstream conditions may be not always possible. (3). Very low interplanetary magnetic field, necessary for high-$\beta$ events, is subjected to strong (in relative terms) intrinsic spatial and temporal variability, thus actual $\beta$ conditions and IMF vector need to be always rechecked with**

[Figure]

**Figure 1.** Number of hours with high $\beta$ with respect to calendar year.

**local measurements. This issue is further illustrated with the event selection results below and is elaborated more in Discussion.**

**Since the high-$\beta$ shocks are rare, it is unreasonable to search for them, rechecking every registered event. It is more practical first to identify the intervals with the suitable conditions of solar wind.** The semi-automated algorithm is used to

5 assemble initial statistics of the shock candidates. For each 1-hour point in OMNI with $\beta > 10$, we check for possible spacecraft location within 5 $R_E$ from the model bow shock (Farris et al., 1991). In a case any spacecraft is in the right place, the plots of solar wind, IMF, local magnetic field and plasma parameters are analyzed visually in the 5-hour window around the selected hour. These broad temporal and spatial spans are used to ensure that all possible crossings of a moving bow shock are captured for future analysis. Only events with the clear shock traversals (jumps in magnetic field and ion density) are accepted. Such a

10 manual selection has definite bias to quasi-perpendicular and oblique shocks (which usually have a clear step-like appearance), but it is considered acceptable for this particular study. The most of these initially selected intervals actually contain no shock crossings.

Discovered particular shock crossings are checked with 1-min OMNI data. Plasma $\beta$ is often below 10, either because registered shocks are just outside of initially selected hours, or because $\beta$ varied on a time scale, smaller than an hour. Since

15 a change of $\beta$ is usually related with the solar wind density change, it is associated also with the dynamic pressure change. The latter drives a large-scale shock motion and probability of shock registration by a spacecraft increases. In fact, many shock crossings are registered at a boundary of $\beta$ change and such events are also discarded, since it isimpossible to attribute them to stable upstream plasma conditions.

Finally the list contained about a hundred individual crossings with average $\beta$ about 20 (taken as 1-min OMNI value at the

20 moment of shock front crossing). **About ten events occurred with very high $\beta > 40$.** The choice of initial threshold $\beta > 10$ (for 1-hour points) was finally justified at this stage, since a variant with initial $\beta > 20$ resulted with the almost empty final list.

[Figure]

**Figure 2.** Histograms of solar wind and IMF occurrence for 1995–2017 (solid lines) and for $\beta > 10$ (dashed lines) subset. (a) Total magnetic field (red line corresponds to $\beta > 20$) , (b) total static pressure, (c) solar wind speed, (d) ion density.

However, all these events still need a more detailed confirmation, in particular, of local high $\beta$, stable enough crossing velocity, plasma data availability etc.

For the specific **multipoint analysis** in this investigation we selected 22 Cluster project crossings with relatively small spacecraft separation. One event is from 2003, with the Cluster tetrahedron size of about 300 km, while the other are for the
5   late years 2008–2016, **when separation only between a pair of Cluster spacecraft C3 and C4 was controlled (30–150 km for our events).** The full list is in Table S1 in Supplement 1. **This uneven annual distribution is a consequence of solar cycle dependence. Events are grouped in 7 days. Specifically, 5 crossings are registered within one hour at December 18, 2011, 4 crossings — within two hours at January 3, 2008, 8 crossings — within two hours at January 4, 2008, 2 crossings — within one hour at February 16, 2012. However not all these adjacent crossings are similar.** Some of these examples
10   are presented below.

[Figure]

**Figure 3.** Example of high-$\beta$ interval. From top to down: magnetic field magnitude, solar wind speed, proton density, proton temperature, plasma $\beta$. 1-min OMNI data set used.

**3 Shock examples**

**3.1 Event 1**

The first example is registered by Cluster C3 and C4 spacecraft on 18 December 2011 (14:36–14:40 UT) with the separation 36 km. The spacecraft orbit is almost parallel to the model shock front (Fig. 5), but shock velocity is definitely much higher than the spacecraft velocity. Solar wind speed is small $\sim$260 km/s, IMF magnitude — 2.5 nT (all characteristics are in Table S1). Alfven Mach number is $\approx$18, magnetosonic Mach number is $\approx$5, $\beta$ (according to 1-min OMNI) is 10.8. **Solar wind magnetic field measured locally by Cluster is the same as OMNI data (compare two lines in Fig. 6d), therefore OMNI $\beta$ value is confirmed. OMNI IMF vector direction is only $\sim 10^o$ different with the local upstream field taken at 14:40–14:41 UT (not shown here). The model shock normal angle with respect to OMNI (local) IMF $\theta_{Bn}$ is 46$^o$ (54$^o$) (using** Farris et al. (1991) model). **The coplanarity calculation for the shock normal results in $\theta_{Bn}$ equal to** 42$^o$**.** Thus this is

[Figure]

**Figure 4.** Comparison of Wind and ACE $\beta$ using 1-hour data. See text for details. Red line is bisector

quasi-perpendicular or oblique supercritical bow shock with the reliably determined geometry. It's structure for more standard $\beta$ is well studied (Scudder et al., 1986; Krasnoselskikh et al., 2013; Lefebvre et al., 2009).

Fig. 6 contains overview of magnetic field and plasma parameters. **The transition lasts about 200 seconds 14:37:00-14:40:30 from the first signs of gyrating ions upstream (Fig. 6f) up to the stable downstream conditions. The increase**
5  **in magnetic field magnitude (aka shock ramp in a quasi-perpendicular case) is smeared within half a minute 14:37:45-14:38:20 UT  and is accompanied with the similar smeared increase of ion density.** The nominal shock front transition is somewhat arbitrarily placed at 14:37:45 UT (marked by vertical line) at a first extended peak of magnetic field. The magnetic field increase is wavy, rather than regular or step-like, magnetic magnitude immediately downstream is often down to 5 nT. Thus it proved to be impossible to determine with multipoint analysis the shock speed. The final value of downstream magnetic
10  field is around 10 nT, and compression ratio is thus close to maximally possible value of 4, in accordance with the high Mach number.

**Despite the described smeared magnetic field increase, the full shock transition is rather compact and coherent and thus it is distinctly different from what expected for quasi-parallel shock with multiple shocklets (Burgess et al., 2005). The smeared increase of magnetic field magnitude may be attributed to relatively large ion gyroradius in low IMF, or**
15  **interpreted as a cyclic reformation, similar to that of oblique shock (Lefebvre et al., 2009). A more detailed phenomeno-logical description of this shock transition requires analysis of ion kinetics, which will be performed elsewhere.**

**We highlight in Figure 7 the interval with the strongest low-frequency magnetic variations.** Frequency spectra are in Figure 8. The magnetic profile is dominated by a variation with frequency around 0.3 Hz and amplitude up to 20 nT, more pronounced in $B_y$. An interval 14:37:27–14:37:47 is taken to estimate the wavelength. The main oscillation (0.3 Hz) is very

similar at two spacecraft and visually the time shift between C3 and C4 is about a fraction of a second. Since the variation has a clear dominating frequency it is more convenient to perform the time-domain multi-point analysis.

The parameters of magnetic variations, filtered in frequency range 0.1–0.77 Hz, are in Table 1. Vector of maximum variance is almost along local magnetic field ($B_y$ component dominates), of minimum variance — along $Z$. Ratios of eigenvalues are $\lambda_{min}/\lambda_{int} = 0.34$, $\lambda_{int}/\lambda_{max} = 0.58$, and one may assume elliptic polarisation. The time shift between magnetic measurements along the maximum variance component, determined with the correlation analysis, is 0.13 s. This value is rather reliably calculated, since it is 2–3 times larger than the sampling interval. The spacecraft separation along the minimum variance direction is 10 km and the resulting wavelength estimate is ∼250 km. However the hodograph of magnetic field rotation (Fig. 9) shows that the polarization actually might be linear with the variable eigenvector (but mostly along the determined maximum variance). In such a case propagation direction cannot be defined with the variance analysis. **Alternatively, for the compressive low frequency MHD waves the propagation direction can be determined with the coplanarity approach (Hubert et al., 1998) (the maximum variance direction, the magnetic field direction and the wavevector should be in the same plane). However, in our case, the angle between the maximum variance direction and the local magnetic field is rather small (only $12^o$) and coplanarity calculation result would be unrealiable.**

We also estimate the span of principally possible wavelengths. The maximal one is ∼900 km, obtained taking full spacecraft separation 36 km. The Doppler shift is 0.04–0.58 Hz, depending on a wavelength and taken local proton velocity value (full 146 km/s or its projection to minimal eigenvector 41 km/s).

Finally we note the oscillations with higher frequency about 1 Hz and smaller amplitude of couple nT, which are best observable in $B_z$ component (Fig. 7c and Fig. 8). The eigenvalue ratios (after filtering the frequency range 0.7–10 Hz) are $\lambda_{min}/\lambda_{int} = 0.68$, $\lambda_{int}/\lambda_{max} = 0.49$, thus reliable determination of any wave proper direction is definitely not possible. Oscillations are quite different at two spacecraft and the multipoint analysis also proved to be not possible.

**3.2 Event 2**

A shock from January 4th, 2008 (16:00–16:04 UT) was registered with Cluster C3 and C4 separation about 40 km. General event parameters are in Table S1, overview of plasma and magnetic field parameters is Fig.S1 in Supplement. The detailed wave activity at the front is presented in Fig. 10. Solar wind parameters and general crossing structure are very similar to that for the Event 1. Solar wind speed is small ∼315 km/s, IMF magnitude — 2.4 nT. Alfven Mach number is ≈23, magnetosonic Mach number is ≈7, current $\beta$ (according to 1-min OMNI) is 12.2. **Solar wind magnetic field measured locally by Cluster is the same as OMNI data (compare two lines in Fig. S1d), therefore OMNI $\beta$ value is confirmed. All variants for $\theta_{Bn}$ give around $40^o$.**

The transition lasts about 2 minutes 16:00:50-16:02:50 from the first signs of gyrating ions upstream to stable downstream conditions (Fig. S1f). The jump in magnetic field magnitude and ion density is smeared within half a minute 16:01:30-16:02:00 UT, and is wavy rather than step-like, downstream magnetic magnitude is often as small as 2–5 nT. The nominal shock front transition is somewhat arbitrarily placed at 16:01:35 UT at a first extended peak of magnetic field.

[Figure]

**Figure 5.** Spacecraft orbit and model shock position for shock 12 December 2011.

**Table 1.** Wave analysis data for shock 18 December 2011, 14:37:27–14:37:47.

| | |
|---|---|
| max eigenvector, $V_{max}$ | -0.23, 0.94, 0.27 |
| med eigenvector, $V_{med}$ | 0.97, 0.20, 0.15 |
| min eigenvector, $V_{min}$ | -0.08, -0.29, 0.95 |
| eigenvalues | 2.23, 6.64, 11.50 |
| magnetic field C3, $B_3$ (nT) | -3.58, 9.53, 0.96 |
| local proton velocity C4 (km/s) | -118.1, 82.1, -29.29 |
| angle, $V_{max}$ and IMF | $34^o$ |
| angle, $V_{min}$ and IMF | $110^o$ |
| angle, $V_{max}$ and $B_3$ | $12^o$ |
| angle, $V_{min}$ and $B_3$ | $99^o$ |
| peak frequency in max component | 0.3 Hz |
| time shift in magnetic field along $V_{max}$ | 0.13 s |
| separation along $V_{min}$ | 10 km |
| wavelength | 252 km |

[Figure]

**Figure 6.** Overview of C4 magnetic and plasma measurements for event 18 December 2011. (a) proton velocity, (b) proton density and OMNI solar wind density, (c) proton parallel and perpendicular temperature, (d) magnetic field magnitude and OMNI IMF magnitude, (e,f) proton spectrograms for the sunward and dawnward looking sectors.

The full resolution waveform is in Figure 10. Similar to Event 1, there is a dominating oscillation with frequency about 0.4–0.5 Hz, as well as the lower amplitude waves with frequency above 1 Hz (Fig. S2). The specific feature is strong difference of C3 and C4 oscillation amplitudes during the first 20 s downstream the front (16:01:25–16:01:35 UT) despite relatively small separation. The presence of such difference in amplitudes is typical for all shocks registered during this day (8 crossings within 2 hours in Table S1).

Nevertheless, it is possible to perform multipoint separation analysis for the interval 16:01:15-16:01:25, where two waveforms in $B_y$ component (Fig. 10b) are very similar and shifted by a fraction of period. All wave parameters (filtered in the range 0.1–2 Hz) are in Table 2. As in Event 1, maximum eigenvector is almost along $Y$, medium eigenvector — along $X$. Ratios of eigenvalues are $\lambda_{min}/\lambda_{int} = 0.15$, $\lambda_{int}/\lambda_{max} \approx 0.5$, thus minimum variance (nominal propagation) direction is well defined. The time shift between the magnetic measurements along the maximum variance component is 0.22 s (determined with correlation analysis), while the spacecraft separation along the minimum variance direction is 6.8 km. The resulting wavelength estimate is 61 km for the peak frequency 0.5 Hz. This value is close to spacecraft

[Figure]

**Figure 7.** Full resolution magnetic waveform for shock 18 December 2011. In panels (a-d) are components and total value of magnetic field.

separation distance (about 40 km) and thus is generally consistent with the nearby observation of substantial difference between magnetic fields at C3 and C4 (at 16:01:25–16:01:35).

The hodograph of magnetic field rotation (Fig. 10), however, shows absence of any stable polarization, which can be interpreted as sometimes linear, sometimes circular. The coplanarity approach again can not be used here to confirm the wave vector direction since the angle between the maximum variance direction and the local magnetic field is rather small (20$^o$). The maximum possible wavelength (if spacecraft separation along wave vector is maximal 40 km) is ∼400 km.

**3.3 Event 3**

One more crossing is from January 3rd, 2008 (14:30–1435 UT) with Cluster separation ∼100 km (Table S1, Fig. S3 in Supplement). OMNI data showed very low IMF (1.1 nT) and $\beta = 39$. Solar wind speed is small ∼321 km/s, Alfven Mach number is ≈42, magnetosonic Mach number is ≈7. The model $\theta_{Bn}$ is 47$^o$. In Fig. 12 we present a view of local magnetic field along with OMNI data. Though local upstream magnitude is approximately equal to that in OMNI (except starting from 14:30 UT closer to the shock), the upstream field direction is changing by more than 90$^o$ and

[Figure]

**Figure 8.** C3 frequency spectra for $B_y$, $B_z$ components and magnetic field magnitude for shock 18 December 2011.

**Table 2.** Wave analysis data for shock 04 January 2008, 16:01:15–16:01:25.

| | |
|---|---|
| max eigenvector, $V_{max}$ | -0.46 0.87 0.17 |
| med eigenvector, $V_{med}$ | 0.88 0.42 0.22 |
| min eigenvector, $V_{min}$ | -0.12 -0.25 0.96 |
| eigenvalues | 3.4, 22.9, 45.3 |
| magnetic field C3, $B_3$ (nT) | -9.05, 9.85, -0.75 |
| local proton velocity C4 (km/s) | -178.3, 125.7, -67.4 |
| angle, $V_{max}$ and IMF | $46^o$ |
| angle, $V_{min}$ and IMF | $79^o$ |
| angle, $V_{max}$ and $B_3$ | $20^o$ |
| angle, $V_{min}$ and $B_3$ | $99^o$ |
| peak frequency in max component | 0.5 Hz |
| time shift in magnetic field along $V_{max}$ | 0.22 s |
| separation along $V_{min}$ | 6.8 km |
| wavelength | 61 km |

the model $\theta_{Bn}$ is also changing to more perpendicular geometry. The presence of an earlier shock crossing at 14:20 UT may also affect observed upstream conditions. Downstream magnetic field is also strongly changing direction with

[Figure]

**Figure 9.** Hodographs of C3 magnetic field in eigenvector coordinates for shock 18 December 2011.

**a temporal scale of about a minute (Fig.12, right side). Therefore for this shock reliable determination of magnetic geometry is impossible. This problem may be inherently related with very small value of upstream magnetic field.**

**Fig. S3 contains overview of magnetic field and plasma parameters. The transition lasts about 2.5 minutes 14:32:00–14:34:30 from the first signs of gyrating ions upstream and growth of parallel ion temperature (Fig. S3e,f) to stable downstream conditions. The jump in magnetic field magnitude is smeared within half a minute 14:34:00–14:34:30 UT, It is wavy rather than step-like and magnetic magnitude downstream is often as small as 1–2 nT.** The nominal shock front transition is somewhat arbitrarily placed at 14:34:10 UT (marked by vertical line in Fig.S3) at a first extended peak of magnetic field. Some increase of variation amplitudes around 14:34:10 can be interpreted as a localized front intensification or as a result of shock bounce motion.

[Figure]

**Figure 10.** Full resolution magnetic waveform for shock 04 January 2008. In panels (a-d) are components and total value of magnetic field

The detailed view of magnetic variations at the front is in Fig. 13. Only relatively high frequency oscillations about 2 Hz are present (frequency spectra are in Fig. S4). There are no wave packets with the stable phase. For example, at 14:34:10–14:34:14 UT $X$ and $Z$ components are in anticorrelation for C3 and C4, while immediately near, at 14:34:08–14:34:10 UT these components are in phase. Therefore, the reliable multipoint analysis for this event is impossible. Magnetic field hodograph plot for 14:34:10–14:34:14 is in Fig.14. It confirms unstable (but consistent with the changing linear) polarization. However, assuming that C3 and C4 variations are mostly in antiphase (half a period between spacecraft), one gets the maximal wavelength estimate ∼**200** km.

**3.4 Observation summary and statistics**

Our statistics includes 22 oblique and quasi-perpendicular shocks. The minimum $\theta_{Bn}$ **37**$^o$, two largest ones are 62 and 83$^o$. Values of $\beta$ range from 39 to 7.5. All cases are supercritical shocks with magnetosonic Mach number more than 5.5. Alfvenic Mach numbers are large because of large $\beta$. All shocks exhibit a clear several-minute-long transition zone between solar wind ion flow and magnetosheath. The main increase of magnetic field and ion density has duration about several tens of seconds. The observed shocks, as concerns their general structure, are typical oblique/quasi-

[Figure]

**Figure 11.** Hodographs of C4 magnetic field in eigenvector coordinates for shock 04 January 2008.

perpendicular shocks, with the somewhat smeared main magnetic field increase. This magnetic profile is typical for all our shocks irrespective of $\theta_{Bn}$ angle.

On a smaller time scales of seconds, the magnetic profile is dominated by very large amplitude magnetic variations, gradually growing in the course of magnetic field increase towards downstream. As a result, the exact location of the 'main' magnetic jump (aka ramp for supercritical quasi-perpendicular shocks) can not be defined. This presence of high-amplitude variations is in agreement with previous publications (Winterhalter and Kivelson, 1988).

The three examples show characteristics of the dominating magnetic variations, typical for all considered events. The detailed multipoint variation analysis allowed to obtain following new information. In the most of shocks (and in Examples 1 and 2) the variations exhibit the well defined frequency peak ∼0.2–0.5 Hz. The magnetic phase portrait of these variations is irregular, with no clear persistent polarization. It can be also interpreted as a linear polarization with

[Figure]

**Figure 12.** Local upstream and OMNI magnetic field for shock 03 January 2008. In panels (a-d) are components and total value of magnetic field

the frequently changing main direction. Such polarization does not allow to determine reliably the wave propagation direction and the wavelength. We get the estimates only in the range several-tens-hundreds km.

Two shock events (Dec. 31, 2003 and our Example 3, Jan. 3, 2008 14:32 UT)) have dominating ∼2 Hz variations, visually with more harmonic waveform, but also with unstable phase. These two shocks are not different from the other events in terms of their general parameters. Moreover one of them (Event 3 above) is registered just 10 min after another crossing, which exhibited the first type of variations. Therefore the presence of '2-Hz' waves might be a signature of some temporal shock front evolution.

**4 Discussion**

**4.1 Reliability of solar wind input**

High-$\beta$ solar wind is relatively rare at the Earth orbit. In our study we accepted somewhat ad-hoc threshold of high $\beta$ equal to 10. Such interplanetary conditions tend to occur during solar minima, being created by slow cold dense solar wind with low

[Figure]

**Figure 13.** Full resolution magnetic waveform for shock 03 January 2008. In panels (a-d) are components and total value of magnetic field

IMF (1–2 nT). However is not always easy to confirm that the observed shock crossing actually occurred in high-$\beta$ solar wind interval, identified in OMNI. The first set of problems is related with association of particular crossings with the stable high-$\beta$. These problems are relatively straightforward to identify and solve. A more substantial problem is related with inherent solar wind and IMF variability. We measure solar wind in L1 halo orbit, 1.5 million kilometers away from Earth and with halo radius

5   not less than 200 000 km (for ACE spacecraft). A substantial part of modern OMNI data are taken from Wind spacecraft, which is currently on a much wider halo orbit (300–400 thousand km) (Podladchikova et al., 2018). Solar wind and IMF structures at L1 are not necessarily the same, that actually affect the magnetosphere. The most questionable is spatial persistence of relatively small changes of IMF from 2 to 1 nT, required for creation of very high-$\beta$ intervals.

     Though the specific analysis of the spatial scales of high-$\beta$ areas in the solar wind was not performed, available reports indi-

10   cate significant potential problems. The ISEE data study suggested that during periods of medium to low variance of magnetic field, magnetic features with scale widths of 20 $R_E$ perpendicular to the IMF may occur (Crooker et al., 1982). Comparison of L1 Wind and near-Earth Interball data for 1996–1999 have shown (Petrukovich et al., 2001), that the IMF structures, associated with geomagnetic storms (with the threshold of IMF $B_z$ GSM below –10 nT during 3 hours) are practically the same in L1

[Figure]

**Figure 14.** Hodographs of C4 magnetic field in eigenvector coordinates for shock 03 January 2008 for 14:34:10-14:34:14.

and the near-Earth orbits. However, about 20–80% of the smaller everyday IMF variations, causing substorms (several nT in magnitude on one-hour scale) are different by more than 25% .

Thus the applicability of very high $\beta$ values in OMNI to a shock study is not automatic. It is not always possible to check solar wind $\beta$ immediately before shock crossing with local spacecraft. A spacecraft needs to probe pristine solar wind and then rapidly cross the shock, or there should be an additional near-Earth solar wind monitor. Magnetic field can be reliably measured with magnetometer (still assuming offset uncertainty of about 0.1 nT). Accuracy of ion density and temperature measurements is more problematic, since at L1 the specialized thoroughly calibrated instruments are used, while with a magnetospheric spacecraft, calibration could be rougher for the specific case of solar wind flow. Assumptions on the helium content and electron temperature, used while OMNI $\beta$ calculations, may also result in some errors. Of course as a side product of such variability, an additional (relative to those found in the OMNI set) high $\beta$ intervals may actually form near the bow shock.

**4.2 Shock properties**

The rather compact large scale structure of the observed shock transitions is similar to that reported for oblique and quasi perpendicular shocks. It is distinctly different from the structure of quasi-parallel shocks, which are extended up to several Earth radii. However, there are some differences with low-$\beta$ shocks on a smaller scale. Duration of main magnetic and ion density increase is several tens of seconds. Magnetic variations appear during this increase and have the amplitude comparable or larger than the background magnetic field, so that there is no 'stable' magnetic structure on the time scale of seconds. In comparison, for supercritical quasi-perpendicular low-$\beta$ shocks one usually defines, starting from the upstream, the prolonged interval of somewhat enhanced density and magnetic field (shock foot, lasting tens of seconds) and the sharp main increase (ramp, lasting seconds). The ramp is often used to determine the shock motion with multipoint measurements, but in our case it is impossible. The increased width of main magnetic jump and its wavy nature might be related with some essential scales in the high-$\beta$ plasma or be a sign of the unstable reforming shock front (e.g., Lefebvre et al., 2009).

A shock example with very low upstream magnetic field about 1 nT exhibits very variable direction of magnetic field both upstream and downstream, complicating definition of shock magnetic geometry. This issue might be of a special interest, since at some (very small) value of magnetic field, it's direction should become unimportant for the shock structure. In particular, there will be no difference between perpendicular and parallel shocks and the shock spatial scale should be defined with some nonmagnetic parameters. Ion escape upstream would be controlled then by some diffusive processes. Thus, one of the main topics for the future studies is to observe in greater detail the dependence of the shock scales on the value of $\beta$, especially for the events with the extreme $\beta$. Our statistics has about 10 events with $\beta$ in the range 40–100, comparable with that expected to galaxy clusters plasma (Donnert et al., 2018).

**4.3 Variation properties**

Observed properties of low-frequency magnetic variations (linear polarization with very high amplitude, substantially changing the total magnetic field)immediately suggest their compressive nature and a strong spatial localization due to absence of any

stable several-periods-long wave packets. Thus observed variations are strongly different from that in low-$\beta$ supercritical events (e.g. Krasnoselskikh et al., 2013), where clear whistler wave packets with elliptic polarization dominate. Observed polarization is also not consistent with the earlier suggested alfven mode (Kennel and Sagdeev, 1967).

Dominating wave mode downstream of the shock front was also addressed in a number of other investigations. Hubert et al. (1989) identified mirror waves comparing magnetic field with fast electron measurements of ISEE project. Balikhin et al. (1997) identified intermediate mode with two-point AMPTE data analysis. Lacombe et al. (1992) suggested for higher-$\beta$ shocks the mirror mode with linear polarization, and successfully used coplanarity assumption to define the wavevector direction. Czaykowska et al. (2001) have shown mostly compressive mirror mode in shocks with $\beta > 1$. Therefore, almost full variety of possible wave mode variants was identified.

However all these studies used several-minute data intervals, often several minutes away from the shock transition (and the highest amplitude waves at it), with a motivation to access the long sets of uniform variations. In the most of cases, the analyzed frequencies were below 0.1 Hz. This approach is different from ours, in which we addressed relatively short intervals of most powerful oscillations. Also all variants of the plasma mode analysis critically depend on reliable determination of the wave propagation (wavevector) direction. It can be done with minimum variance analysis in the case of elliptic polarization or with the coplanarity supposition. Unfortunately in our case it proved to be impossible to determine the wavevector direction reliably by both methods. It should be noted that a linearly polarized wave with high amplitude comparable or larger than the background field inevitably has the maximum amplitude direction almost parallel to the main magnetic field, precluding the use of the coplanarity conjecture. The wavelength can be determined independently to wave propagation direction with four closely spaced spacecraft (with separation much smaller than that of Cluster).

An additional wave mode candidate for the high-$\beta$ plasma is a Weibel mode. Pokhotelov and Balikhin (2012) suggested that Weibel mode grows in the finite magnetic field as a mix of two opposite circular polarizations. This variant may be consistent with the observed irregular variations.

**5   Conclusions**

High-$\beta$ ($\beta > 10$) shocks are relatively rare and largely unexplored class of Earth bow shock. Formation of high-$\beta$ interplanetary plasmas is mostly related with dense slow solar wind and very low magnetic field up to 1–2 nT. The higher is $\beta$ (in OMNI), it is more difficult to confirm it locally. Our shock analysis was limited to oblique and quasi-perpendicular cases and shows some differences from supercritical shocks with lower $\beta$ both in the general structure of the shock transition and in the properties of magnetic variations. Magnetic field and ion density jumps are smeared to a couple tens of seconds. Dominating magnetic waves have frequencies 0.2–0.5, sometimes, $\sim$2 Hz, irregular, close to linear, polarization and spatial scales around a hundred km. Recent Magnetospheric Multiscale mission observations with very closely spaced spacecraft are necessary to conclude more definitely on the wave mode.

*Author contributions.* OMC and PIS performed the data processing and analysis. AAP is responsible for data analysis and interpretation. AAP prepared the manuscript with contributions from all co-authors.

*Competing interests.* The authors declare that they have no conflict of interest.

*Acknowledgements.* The data analysis was funded with Russian Science Fund project 05-14-00824. We are thankful for Cluster Science Archive, CDAWeb and OMNI for availability of spacecraft data.

**References**

Axford, W. I., Leer, E., and Skadron, G.: The Acceleration of Cosmic Rays by Shock Waves, International Cosmic Ray Conference, 11, 132, 1977.

Balikhin, M., Gedalin, M., and Petrukovich, A.: New mechanism for electron heating in shocks., , 70, 1259–1262, https://doi.org/10.1103/PhysRevLett.70.1259, 1993.

Balikhin, M. A., Woolliscroft, L. J. C., Alleyne, H. S. C., Dunlop, M., and Gedalin, M. A.: Determination of the dispersion of low frequency waves downstream of a quasiperpendicular collisionless shock, Annales Geophysicae, 15, 143–151, https://doi.org/10.1007/s00585-997-0143-x, 1997.

Balogh, A., Carr, C. M., Acuña, M. H., Dunlop, M. W., Beek, T. J., Brown, P., Fornaçon, K. H., Georgescu, E., Glassmeier, K. H., Harris, J., Musmann, G., Oddy, T., and Schwingenschuh, K.: The Cluster Magnetic Field Investigation: overview of in-flight performance and initial results, Annales Geophysicae, 19, 1207–1217, https://doi.org/10.5194/angeo-19-1207-2001, 2001.

Burgess, D., Lucek, E. A., Scholer, M., Bale, S. D., Balikhin, M. A., Balogh, A., Horbury, T. S., Krasnoselskikh, V. V., Kucharek, H., Lembège, B., Möbius, E., Schwartz, S. J., Thomsen, M. F., and Walker, S. N.: Quasi-parallel Shock Structure and Processes, , 118, 205–222, https://doi.org/10.1007/s11214-005-3832-3, 2005.

Crooker, N. U., Siscoe, G. L., Russell, C. T., and Smith, E. J.: Factors controlling degree of correlation between ISEE 1 and ISEE 3 interplanetary magnetic field measurements, Journal of Geophysical Research, 87, 2224–2230, https://doi.org/10.1029/JA087iA04p02224, 1982.

Czaykowska, A., Bauer, T. M., Treumann, R. A., and Baumjohann, W.: Magnetic field fluctuations across the Earth's bow shock, Annales Geophysicae, 19, 275–287, https://doi.org/10.5194/angeo-19-275-2001, 2001.

Dimmock, A. P., Balikhin, M. A., Walker, S. N., and Pope, S. A.: Dispersion of low frequency plasma waves upstream of the quasi-perpendicular terrestrial bow shock, Annales Geophysicae, 31, 1387–1395, https://doi.org/10.5194/angeo-31-1387-2013, 2013.

Donnert, J., Vazza, F., Brüggen, M., and ZuHone, J.: Magnetic Field Amplification in Galaxy Clusters and Its Simulation, , 214, 122, https://doi.org/10.1007/s11214-018-0556-8, 2018.

Farris, M. H., Petrinec, S. M., and Russell, C. T.: The thickness of the magnetosheath: Constraints on the polytropic index, Geophysical Research Letters, 18, 1821–1824, https://doi.org/10.1029/91GL02090, 1991.

Farris, M. H., Russell, C. T., Thomsen, M. F., and Gosling, J. T.: ISEE 1 and 2 observations of the high beta shock, Journal of Geophysical Research, 97, 19 121–19 127, https://doi.org/10.1029/92JA01976, 1992.

Formisano, V., Russell, C. T., Means, J. D., Greenstadt, E. W., Scarf, F. L., and Neugebauter, M.: Collisionless shock waves in space: A very high $\beta$ structure, Journal of Geophysical Research, 80, 2013, https://doi.org/10.1029/JA080i016p02013, 1975.

Hobara, Y., Balikhin, M., Krasnoselskikh, V., Gedalin, M., and Yamagishi, H.: Statistical study of the quasi-perpendicular shock ramp widths, Journal of Geophysical Research (Space Physics), 115, A11106, https://doi.org/10.1029/2010JA015659, 2010.

Hubert, D., Perche, C., Harvey, C. C., Lacombe, C., and Russell, C. T.: Observation of mirror waves downstream of a quasi-perpendicular shock, Geophysical Research Letters, 16, 159–162, https://doi.org/10.1029/GL016i002p00159, 1989.

Hubert, D., Lacombe, C., Harvey, C. C., Moncuquet, M., Russell, C. T., and Thomsen, M. F.: Nature, properties, and origin of low-frequency waves from an oblique shock to the inner magnetosheath, Journal of Geophysical Research, 103, 26 783–26 798, https://doi.org/10.1029/98JA01011, 1998.

Kennel, C. F. and Sagdeev, R. Z.: Collisionless shock waves in high $\beta$ plasmas: 1, Journal of Geophysical Research, 72, 3303–3326, https://doi.org/10.1029/JZ072i013p03303, 1967.

Kennel, C. F., Edmiston, J. P., and Hada, T.: A quarter century of collisionless shock research, Washington DC American Geophysical Union Geophysical Monograph Series, 34, 1–36, https://doi.org/10.1029/GM034p0001, 1985.

5 Krasnoselskikh, V., Balikhin, M., Walker, S. N., Schwartz, S., Sundkvist, D., Lobzin, V., Gedalin, M., Bale, S. D., Mozer, F., Soucek, J., Hobara, Y., and Comisel, H.: The Dynamic Quasiperpendicular Shock: Cluster Discoveries, , 178, 535–598, https://doi.org/10.1007/s11214-013-9972-y, 2013.

Krasnoselskikh, V. V., Lembège, B., Savoini, P., and Lobzin, V. V.: Nonstationarity of strong collisionless quasiperpendicular shocks: Theory and full particle numerical simulations, Physics of Plasmas, 9, 1192–1209, https://doi.org/10.1063/1.1457465, 2002.

10 Krymskii, G. F.: A regular mechanism for the acceleration of charged particles on the front of a shock wave, Soviet Physics Doklady, 22, 327, 1977.

Lacombe, C., Pantellini, F. G. E., Hubert, D., Harvey, C. C., Mangeney, A., Belmont, G., and Russell, C. T.: Mirror and Alfvenic waves observed by ISEE 1-2 during crossings of the earth's bow shock, Annales Geophysicae, 10, 772–784, 1992.

Lefebvre, B., Seki, Y., Schwartz, S. J., Mazelle, C., and Lucek, E. A.: Reformation of an oblique shock observed by Cluster, Journal of
15 Geophysical Research (Space Physics), 114, A11107, https://doi.org/10.1029/2009JA014268, 2009.

Markevitch, M. and Vikhlinin, A.: Shocks and cold fronts in galaxy clusters, , 443, 1–53, https://doi.org/10.1016/j.physrep.2007.01.001, 2007.

Petrukovich, A. A., Romanov, S. A., and Klimov, S. L.: Direct Measurements of AC Plasma Currents in the Outer Magnetosphere, Washington DC American Geophysical Union Geophysical Monograph Series, 103, 199, https://doi.org/10.1029/GM103p0199, 1998.

20 Petrukovich, A. A., Klimov, S. I., Lazarus, A., and Lepping, R. P.: Comparison of the solar wind energy input to the magnetosphere measured by Wind and Interball-1, Journal of Atmospheric and Solar-Terrestrial Physics, 63, 1643–1647, https://doi.org/10.1016/S1364-6826(01)00039-6, 2001.

Podladchikova, T., Petrukovich, A., and Yermolaev, Y.: Geomagnetic storm forecasting service StormFocus: 5 years online, Journal of Space Weather and Space Climate, 8, A22, https://doi.org/10.1051/swsc/2018017, 2018.

25 Pokhotelov, O. A. and Balikhin, M. A.: Weibel instability in a plasma with nonzero external magnetic field, Annales Geophysicae, 30, 1051–1054, https://doi.org/10.5194/angeo-30-1051-2012, 2012.

Rème, H., Aoustin, C., Bosqued, J. M., Dandouras, I., Lavraud, B., Sauvaud, J. A., Barthe, A., Bouyssou, J., Camus, T., Coeur-Joly, O., Cros, A., Cuvilo, J., Ducay, F., Garbarowitz, Y., Medale, J. L., Penou, E., Perrier, H., Romefort, D., Rouzaud, J., Vallat, C., Alcaydé, D., Jacquey, C., Mazelle, C., D'Uston, C., Möbius, E., Kistler, L. M., Crocker, K., Granoff, M., Mouikis, C., Popecki, M., Vosbury, M.,
30 Klecker, B., Hovestadt, D., Kucharek, H., Kuenneth, E., Paschmann, G., Scholer, M., Sckopke, N., Seidenschwang, E., Carlson, C. W., Curtis, D. W., Ingraham, C., Lin, R. P., McFadden, J. P., Parks, G. K., Phan, T., Formisano, V., Amata, E., Bavassano- Cattaneo, M. B., Baldetti, P., Bruno, R., Chionchio, G., di Lellis, A., Marcucci, M. F., Pallocchia, G., Korth, A., Daly, P. W., Graeve, B., Rosenbauer, H., Vasyliunas, V., McCarthy, M., Wilber, M., Eliasson, L., Lundin, R., Olsen, S., Shelley, E. G., Fuselier, S., Ghielmetti, A. G., Lennartsson, W., Escoubet, C. P., Balsiger, H., Friedel, R., Cao, J. B., Kovrazhkin, R. A., Papamastorakis, I., Pellat, R., Scudder, J., and Sonnerup,
35 B.: First multispacecraft ion measurements in and near the Earth's magnetosphere with the identical Cluster ion spectrometry (CIS) experiment, Annales Geophysicae, 19, 1303–1354, https://doi.org/10.5194/angeo-19-1303-2001, 2001.

Sagdeev, R. Z.: Cooperative Phenomena and Shock Waves in Collisionless Plasmas, Reviews of Plasma Physics, 4, 23, 1966.

Schwartz, S. J., Henley, E., Mitchell, J., and Krasnoselskikh, V.: Electron Temperature Gradient Scale at Collisionless Shocks, , 107, 215002, https://doi.org/10.1103/PhysRevLett.107.215002, 2011.

Scudder, J. D., Mangeney, A., Lacombe, C., Harvey, C. C., Aggson, T. L., Anderson, R. R., Gosling, J. T., Paschmann, G., and Russell, C. T.: The resolved layer of a collisionless, high $\beta$, supercritical, quasi-perpendicular shock wave 1. Rankine- Hugoniot geometry, currents, and stationarity, Journal of Geophysical Research, 91, 11 019–11 052, https://doi.org/10.1029/JA091iA10p11019, 1986.

Vasko, I. Y., Mozer, F. S., Krasnoselskikh, V. V., Artemyev, A. V., Agapitov, O. V., Bale, S. D., Avanov, L., Ergun, R., Giles, B., Lindqvist, P. A., Russell, C. T., Strangeway, R., and Torbert, R.: Solitary Waves Across Supercritical Quasi-Perpendicular Shocks, Geophysical Research Letters, 45, 5809–5817, https://doi.org/10.1029/2018GL077835, 2018.

Walker, S., Alleyne, H., Balikhin, M., André, M., and Horbury, T.: Electric field scales at quasi-perpendicular shocks, Annales Geophysicae, 22, 2291–2300, https://doi.org/10.5194/angeo-22-2291-2004, 2004.

Wilson, Lynn B., I., Stevens, M. L., Kasper, J. C., Klein, K. G., Maruca, B. A., Bale, S. D., Bowen, T. A., Pulupa, M. P., and Salem, C. S.: The Statistical Properties of Solar Wind Temperature Parameters Near 1 au, The Astrophysical Journal Supplement Series, 236, 41, https://doi.org/10.3847/1538-4365/aab71c, 2018.

Winterhalter, D. and Kivelson, M. G.: Observations of the Earth's bow shock under high Mach number/high plasma beta solar wind conditions, Geophysical Research Letters, 15, 1161–1164, https://doi.org/10.1029/GL015i010p01161, 1988.